# Action Dependency Graphs for Globally Optimal Coordinated Reinforcement Learning

## Abstract

Action-dependent policies, which condition decisions on both states and other agents' actions, provide a powerful alternative to independent policies in multi-agent reinforcement learning. Most existing studies have focused on auto-regressive formulations, where each agent's policy depends on the actions of all preceding agents. However, this approach suffers from severe scalability limitations as the number of agents grows. In contrast, sparse dependency structures, where each agent relies only on a subset of other agents, remain largely unexplored and lack rigorous theoretical foundations. To address this gap, we introduce the action dependency graph (ADG) to model sparse inter-agent action dependencies. We prove that action-dependent policies can converge to solutions stronger than Nash equilibria, which often trap independent policies, and we refer to such solutions as $G_d$-locally optimal policies. Furthermore, within coordination graph (CG) structured problems, we show that a $G_d$-locally optimal policy attains global optimality when the ADG satisfies specific CG-induced conditions. To substantiate our theory, we develop a tabular policy iteration algorithm that converges exactly as predicted. We further extend a standard deep MARL method to incorporate action-dependent policies, confirming the practical relevance of our framework.

## 1 Introduction

Achieving effective multi-agent reinforcement learning (MARL) in fully cooperative environments requires agents to coordinate their actions to maximize collective performance. Most existing MARL methods rely on independent policies (Zhang et al., 2021; Oroojlooy & Hajinezhad, 2023), where each agent makes decisions based solely on its state or observation. Although computationally tractable and scalable, these completely decentralized policies are often suboptimal (Fu et al., 2022). The primary limitation lies in their tendency to converge to one of many Nash equilibrium solutions (Ye et al., 2022), which may not correspond to the globally optimal solution.

The emergence of action-dependent policies (Fu et al., 2022) offers a promising solution to this challenge. By incorporating the actions of other agents into an agent's decision-making process, action-dependent policies enable more effective cooperation and achieve superior performance compared to independent policies. We introduce the action dependency graph (ADG), a directed acyclic graph, to represent the action dependencies required for agents to make decisions. Theoretical studies (Bertsekas, 2021; Chen & Zhang, 2023) demonstrate that policies with auto-regressive forms, associated with fully dense ADGs—where replacing each directed edge with an undirected edge yields a complete graph—guarantee global optimality. However, fully dense ADGs pose substantial scalability issues, as they require a high degree of interdependence and coordination.

Sparse ADGs, which involve fewer inter-agent dependencies, offer a more scalable alternative. This leads to a critical question: can action-dependent policies with sparse ADGs still guarantee global optimality? To answer this question, we build on the framework of coordinated reinforcement learning (Guestrin et al., 2002), where the cooperative relationship between agents is described by a coordination graph (CG). We find that global optimality can still be achieved using an action-dependent policy with a sparse ADG, provided that a specific relationship between the ADG and the CG is satisfied.

The contributions of this paper are summarized as follows. (i) We introduce the notion of a $G_d$-locally optimal policy, which differs from the Nash equilibrium and more precisely characterizes

the convergence behavior of action-dependent policies. (ii) We establish a theoretical framework that unifies coordination graphs with action-dependent policies and derive optimality conditions for sparse ADGs. To the best of our knowledge, this is the first work to seamlessly integrate these two perspectives. (iii) We design a policy iteration algorithm that, grounded in our theory, guarantees convergence of action-dependent policies to a $G_d$-locally optimal policy, and further to a globally optimal policy under the optimality conditions.

## 2 RELATED WORK

**Independent policy.** The majority of the literature on MARL represents the joint policy as the Cartesian product of independent individual policies. Value-based methods such as IQL (Tan, 1993), VDN (Sunehag et al., 2018), QMIX (Rashid et al., 2018), and QTRAN (Son et al., 2019) employ local value functions that depend only on the state or observation of each agent. Similarly, policy-based methods such as MADDPG (Lowe et al., 2017), COMA (Foerster et al., 2018), MAAC (Iqbal & Sha, 2019), and MAPPO (Yu et al., 2022) directly adopt independent policies. These approaches often fail to achieve global optimality, as they are not able to cover all strategy modes (Fu et al., 2022).

**Coordination graph.** Some value-based methods (Böhmer et al., 2020; Castellini et al., 2021; Li et al., 2021; Wang et al., 2022b) recognize that the limitation of independent policies is due to a game-theoretic pathology known as relative overgeneralization (Panait et al., 2006). To mitigate this, they employ a higher-order value decomposition framework by introducing the coordination graph (CG) (Guestrin et al., 2002). In this graph, the vertices represent agents, and the edges correspond to pairwise interactions between agents in the local value functions. While CGs improve cooperation by considering inter-agent dependencies, the resulting joint policy cannot be decomposed into individual policies. Consequently, decision-making algorithms still require intensive computation, such as Max-Plus (Rogers et al., 2011) or Variable Elimination (VE) (Bertele & Brioschi, 1972). When the CG is dense, these computations may become prohibitively time consuming, making the policy difficult to execute in real time.

**Action-dependent policy.** In contrast to independent policies, action-dependent policies (Wang et al., 2022a; Ruan et al., 2022; Li et al., 2023; 2024) incorporate not only the state, but also the actions of other agents into an agent's decision-making process. The action dependencies among agents can be represented by a directed acyclic graph, which we refer to as the action dependency graph (ADG). In some literature, the action-dependent policy is also referred to as Bayesian policy (Chen & Zhang, 2023) or auto-regressive policy (Fu et al., 2022). Moreover, the use of action-dependent policies can be viewed as a mechanism to leverage communications for enhancing cooperation (Zhou et al., 2023; Duan et al., 2024; Jing et al., 2024). Some approaches (Bertsekas, 2021; Ye et al., 2022; Wen et al., 2022) transform a multi-agent MDP into a single-agent MDP with a sequential structure, enabling each agent to consider the actions of all previously decided agents during decision-making. This transformation ensures the convergent joint policy to be globally optimal (Bertsekas, 2021). However, the fully dense ADG makes these methods computationally expensive and impractical for large-scale systems. For more general ADGs, existing theories can only guarantee convergence to a Nash equilibrium solution (Chen & Zhang, 2023). Currently, no theoretical evidence demonstrates the superiority of action-dependent policies with sparse dependency graphs over independent policies.

## 3 PRELIMINARY

We formulate the cooperative multi-agent reinforcement learning problem as a *Multi-Agent Markov Decision Process* (MAMDP), represented by the tuple $\langle \mathcal{N}, \mathcal{S}, \mathcal{A}, P, r, \gamma \rangle$, where $\mathcal{N} = \{1, \ldots, n\}$ denotes the set of agents, $\mathcal{S}$ is the finite state space, $\mathcal{A} = \prod_{i=1}^{n} \mathcal{A}_i$ is the joint action space formed by the Cartesian product of each agent's finite action space, $P : \mathcal{S} \times \mathcal{A} \times \mathcal{S} \to [0, 1]$ is the transition kernel, $r : \mathcal{S} \times \mathcal{A} \to \mathbb{R}$ is the reward function, and $\gamma \in [0, 1)$ is the discount factor.

We consider policies of the deterministic form $\pi : \mathcal{S} \to \mathcal{A}$. The state value function and state-action value function induced by a policy $\pi$ are

$$V^\pi(s) := \mathbb{E}_\pi \left[ \sum_{t=0}^{\infty} \gamma^t r(s^t, a^t) \,\middle|\, s^0 = s \right], Q^\pi(s, a) := \mathbb{E}_\pi \left[ \sum_{t=0}^{\infty} \gamma^t r(s^t, a^t) \,\middle|\, s^0 = s, a^0 = a \right], \quad (1)$$

where the expectation $\mathbb{E}$ is taken over all random variables $s^t$ induced by $\pi$ and $P$. For any function $V \in \mathcal{R}(\mathcal{S})$, where $\mathcal{R}(\mathcal{S})$ denotes the set of real-valued functions $J : \mathcal{S} \to \mathbb{R}$, we define

$$Q^V(s,a) := r(s,a) + \gamma \sum_{s' \in \mathcal{S}} P(s'|s,a)V(s'). \tag{2}$$

The Bellman operator $\mathcal{T}_\pi : \mathcal{R}(\mathcal{S}) \to \mathcal{R}(\mathcal{S})$ and the Bellman optimality operator $\mathcal{T} : \mathcal{R}(\mathcal{S}) \to \mathcal{R}(\mathcal{S})$ are given by

$$\mathcal{T}_\pi V(s) = Q^V(s, \pi(s)), \quad \mathcal{T}V(s) = \max_{a \in \mathcal{A}} Q^V(s,a). \tag{3}$$

The value function $V^\pi$ is the unique fixed point of $\mathcal{T}_\pi$, and the optimal value function $V^*$ is the unique fixed point of $\mathcal{T}$.

### 3.1 COORDINATION GRAPH

In many practical scenarios such as sensor networks (Zhang & Lesser, 2011), wind farms (Bargiacchi et al., 2018), mobile networks (Bouton et al., 2021), etc., the $Q$-function can be approximated as the sum of local value functions, each depending on the states and actions of a subset of agents. A widely used approach to representing this decomposition is the use of the *coordination graph* (CG) (Guestrin et al., 2002), which captures the pairwise coordination relationships between agents. Formally, we define a CG as follows.

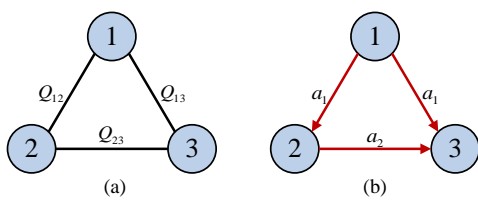

Figure 1: A coordination graph (a) and an action dependency graph (b).

**Definition 3.1** (Coordination Graph). An undirected graph[1] $G_c = (\mathcal{N}, \mathcal{E}_c)$ is a CG under state $s \in \mathcal{S}$ of a value function $Q^\pi : \mathcal{S} \times \mathcal{A} \to \mathbb{R}$, if there exists a local value function $Q_{ij}^\pi : \mathcal{S} \times \mathcal{A}_i \times \mathcal{A}_j \to \mathbb{R}$ for every edge $(i,j) \in \mathcal{E}_c$, and a local value function $Q_i^\pi : \mathcal{S} \times \mathcal{A}_i \to \mathbb{R}$ for every vertex $i \in \mathcal{N}$, such that for any $a \in \mathcal{A}$, the following decomposition holds:

$$Q^\pi(s,a) = \sum_{i \in \mathcal{N}} Q_i^\pi(s, a_i) + \sum_{(i,j) \in \mathcal{E}_c} Q_{ij}^\pi(s, a_i, a_j). \tag{4}$$

*Remark* 3.2. If $G_c$ is a subgraph of $G_c'$, and $G_c$ is a CG of $Q^\pi$, then $G_c'$ is also a CG of $Q^\pi$. Therefore, multiple CGs may correspond to the same value function $Q^\pi$.

Without loss of generality, we assume that $G_c$ is connected; otherwise, the problem can be decomposed into independent subproblems depending on the connected components of $G_c$. In a connected graph, each vertex is involved in an edge, allowing the local value functions associated with vertices to be merged into those local value functions associated with edges, yielding:

$$Q^\pi(s,a) = \sum_{(i,j) \in \mathcal{E}_c} Q_{ij}^\pi(s, a_i, a_j). \tag{5}$$

Figure 1 (a) shows a CG where $Q^\pi$ can be decomposed as:

$$Q^\pi(s,a) = Q_{12}^\pi(s, a_1, a_2) + Q_{13}^\pi(s, a_1, a_3) + Q_{23}^\pi(s, a_2, a_3). \tag{6}$$

Throughout this paper, we focus on a MAMDP structured by a CG.

### 3.2 NOTATIONS

In the paper, we frequently use sets as subscripts in expressions. Let $S \subseteq \mathcal{N}$, and denote its elements in ascending order as $S = \{s_1, s_2, \ldots, s_k\}$. For a space, such as $\mathcal{A}_S$, we define $\mathcal{A}_S := \prod_{i \in S} \mathcal{A}_i := \prod_{i=1}^k \mathcal{A}_{s_i}$. For a vector, such as $a_S$, we define $a_S := (a_{s_1}, a_{s_2}, \ldots, a_{s_k})$, $a_{s_i} \in \mathcal{A}_{s_i}$. The notation $< i$ indicates the set of agents with indices smaller than $i$, similarly for $\leq i$, $> i$, and $\geq i$. For an undirected graph $G_c$, $N_{G_c}(i)$ denotes the neighbor of vertex $i$. When there is no ambiguity, we abbreviate $N_{G_c}(i)$ as $N_c(i)$. $N_c[i] := N_c(i) \cup i$, and $N_c(S)$ denotes the neighbors of a set $S$, that is, $N_c(S) = \bigcup_{i \in S} N_c(i) \setminus S$. For a directed graph $G_d$, $N_{G_d}(i)$ denotes the parent set of vertex $i$. Likewise, we abbreviate $N_{G_d}(i)$ as $N_d(i)$. Similarly, $N_d[i] := N_d(i) \cup i$, and $N_d(S)$ denotes the set of all parent nodes of vertices in $S$.

---

[1]In this paper, the vertices and edges of the graph are represented by agent indices and index pairs.

## 4 ADG WITH OPTIMALITY GUARANTEE

### 4.1 ACTION DEPENDENCY GRAPH

In MARL, a deterministic joint policy $\pi(s)$ is a vector-valued mapping from states to actions, where each component corresponds to an individual action. Thus, $\pi(s)$ can always be written as a collection of independent policies $(\pi_1(s), \pi_2(s), \ldots, \pi_n(s))$. In this sense, the independent policies is expressive enough to represent any joint deterministic policy, including an optimal one. However, this does not imply that independent learning can converge to the optimal joint policy, since independent policies cannot capture coordinated behaviors that rely on correlated actions across agents. The absence of such correlations may cause independent learners to converge to suboptimal points.

To address this limitation, we introduce a broader class of policies, termed *action-dependent policies*, whose inputs include not only the state but also the actions of other agents. Formally, specifying action-dependent policies requires determining the order in which actions are generated. Without loss of generality, we assume that actions are output according to agent indices. In this case, the general form of agent $i$'s policy is $\pi_i : \mathcal{S} \times \mathcal{A}_{<i} \to \mathcal{A}_i$. Since some policies may not depend on the actions of all preceding agents, we use a *dependency set* to represent this sparse dependency relation.

**Definition 4.1** (Dependency Set). Let $C \subseteq (< i)$ be the dependency set of agent $i$'s policy $\pi_i$ under state $s \in \mathcal{S}$. Then, for any $a_C \in \mathcal{A}_C, a'_{(<i)\backslash C}, a_{(<i)\backslash C} \in \mathcal{A}_{(<i)\backslash C}$, it holds that

$$\pi_i(s, a_C, a'_{(<i)\backslash C}) = \pi_i(s, a_C, a_{(<i)\backslash C}).$$

If $C$ is the dependency set of agent $i$, then the policy of agent $i$ depends only on the actions of agents in $C$. For convenience, depending on the context, we sometimes write $\pi_i(s, a_{<i})$ as $\pi_i(s, a_C)$. For a joint policy, the overall action dependency structure can be represented as a directed acyclic graph (DAG).

**Definition 4.2** (Action Dependency Graph (ADG)). The DAG $G_d = (\mathcal{N}, \mathcal{E}_d)$ is the Action Dependency Graph of the joint policy $\pi$ under state $s \in \mathcal{S}$ if, for any $i \in \mathcal{N}$, $N_d(i)$ is the dependency set of $\pi_i$ under state $s$.

The acyclic nature of the ADG guarantees that dependencies do not form cycles, which would otherwise cause decision-making deadlocks and render the policy infeasible. Figure 1(b) illustrates the ADG of a joint policy $\pi$ with the following form:

$$\pi(s) = (\pi_1(s), \pi_2(s, \pi_1(s)), \pi_3(s, \pi_2(s, \pi_1(s)), \pi_1(s))). \tag{7}$$

From (7), it is evident that expressing the components of a joint policy directly in terms of action-dependent policies becomes cumbersome. To streamline such representations, we recursively define the following policy notation:

$$\pi_{i,C}(s, a_C) = \begin{cases} a_i & \text{if } i \in C, \\ \pi_i(s, \pi_{<i,C}(s, a_C)) & \text{otherwise,} \end{cases} \tag{8}$$

where $\pi_{<i,C} = (\pi_{1,C}, \ldots, \pi_{i-1,C})$. The key difference between $\pi_{i,C}$ and action-dependent policy $\pi_i$ is that the former's actions in $(< i) \backslash C$ are already determined by $\pi_{<i,C}$, and therefore $\pi_{i,C}$ depends only on the actions in $C$. A special case is $\pi_{i,\varnothing}$, where no other agent's action is involved. In this case, $\pi_{i,\varnothing}$ is exactly the standard independent policy, and (7) can be rewritten concisely as

$$\pi(s) = (\pi_{1,\varnothing}(s), \pi_{2,\varnothing}(s), \pi_{3,\varnothing}(s)). \tag{9}$$

### 4.2 COORDINATION POLYMATRIX GAME

A key reason why independent policies often converge to locally optimal solutions is the existence of Nash equilibrium policies (Zhang et al., 2022; Kuba et al., 2022), also known as agent-by-agent optimal policies (Bertsekas, 2021). In this subsection, we illustrate the suboptimality of Nash equilibria through an example of *coordination polymatrix game* (Cai & Daskalakis, 2011), and demonstrate how action-dependent policies can overcome this limitation.

A coordination polymatrix game can be viewed as a single-step decision problem, formulated by a MAMDP tuple $\langle \mathcal{N}, \mathcal{S}, \mathcal{A}, P, r, \gamma \rangle$, with $\mathcal{S} = \varnothing$ and $\gamma = 0$. In addition, the game is equipped

with an undirected graph $G_c = (\mathcal{N}, \mathcal{E}_c)$ and a set of pairwise payoff functions $\{r_{ij}\}_{(i,j)\in\mathcal{E}_c}$, which together determine the global reward $r(a) = \sum_{(i,j)\in\mathcal{E}_c} r_{ij}(a_i, a_j)$. In this setting, $r$ is equivalent to the (state)-action value function $Q : \mathcal{A} \to \mathbb{R}$, and $G_c$ serves as the CG of $Q$. Figure 2 illustrates a polymatrix game with three agents, each having two possible actions, $\mathcal{A}_i = \{0, 1\}, i = 1, 2, 3$. The payoff matrices specify the reward for each agent pair; for example, if agents 1 and 2 both choose action 0, they receive a payoff of 1 together.

For independent policies, the joint policies $\pi = (1, 1, 1)$ and $\pi = (0, 0, 0)$ are both Nash equilibria. However, only $\pi = (0, 0, 0)$ is globally optimal. Although $\pi = (1, 1, 1)$ is suboptimal, no single agent has an incentive to deviate unilaterally, since any individual deviation reduces the total reward.

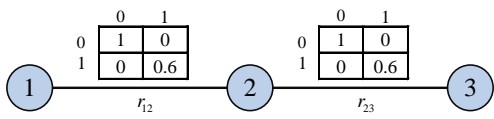

Figure 2: A polymatrix game on a line CG.

Now consider action-dependent policies with an ADG $G_d$ whose edge set is $\mathcal{E}_d = \{(1, 2), (2, 3)\}$. Suppose the policies are given by $\pi_1 = 1$, $\pi_2(0) = 0$, $\pi_2(1) = 1$, $\pi_3(0) = 0$, and $\pi_3(1) = 1$. This corresponds to the same joint policy $\pi = (1, 1, 1)$. However, if agent 1 switches its action to 0, agents 2 and 3 will also switch to 0, leading to the joint action $(0, 0, 0)$ with reward $r(0, 0, 0) = 2$, which exceeds $r(1, 1, 1) = 1.2$. Thus, agent 1 is incentivized to choose action 0, driving the system toward the globally optimal policy.

### 4.3 Optimality Guarantee

The coordination polymatrix game example demonstrates that action-dependent policies can converge to solutions stronger than Nash equilibria. Such solutions are relatively rare in the policy space and are therefore more likely to be globally optimal. We refer to them as $G_d$-locally optimal policies.

**Definition 4.3** ($G_d$-locally Optimal). Let $G_d = (\mathcal{N}, \mathcal{E}_d)$ be a DAG. A joint policy $\pi$ is $G_d$-locally optimal under $s \in \mathcal{S}$ if, for any $a_{N_d(i)} \in \mathcal{A}_{N_d(i)}$, the following holds:

$$Q^\pi(s, \pi_{\mathcal{N}, N_d(i)}(s, a_{N_d(i)})) = \max_{a_i \in \mathcal{A}_i} Q^\pi(s, \pi_{(<i), N_d(i)}(s, a_{N_d(i)}), a_i, \pi_{(>i), N_d[i]}(s, a_{N_d[i]})). \quad (10)$$

When $G_d$ is empty (i.e., independent policies), the notion of $G_d$-local optimality coincides with agent-by-agent optimality. As more edges are added to $G_d$, condition (10) becomes increasingly restrictive. In the extreme case where $G_d$ is a fully dense DAG with edge set $\mathcal{E}_d = \{(i, j) \in \mathcal{N} \times \mathcal{N} : i < j\}$, a $G_d$-locally optimal policy aligns with the globally optimal policy. In general, a joint policy with ADG $G_d$ tends to converge to a $G_d$-locally optimal solution, as discussed in the next section. Thus, the most straightforward way to avoid subopti-

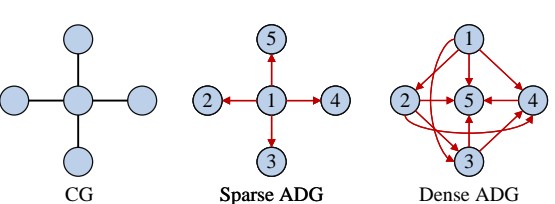

Figure 3: Different index orders of agents result in different sparsity of the ADG.

mality is to adopt a fully dense ADG. However, the computational cost of training and executing such policies grows rapidly with the number of agents, limiting scalability. In fact, if the CG structure can be exploited, even some sparse ADGs suffice to guarantee global optimality. We now introduce a graph condition that links the CG and ADG, ensuring that every $G_d$-locally optimal policy is also globally optimal.

**Theorem 4.4** (Optimality of ADG, proof in Appendix C). *Let $s \in \mathcal{S}$, and let $G_d(s)$ be a DAG and $G_c(s)$ be the CG of $Q^\pi$ under state $s$. Suppose that for every $s \in \mathcal{S}$, the policy $\pi$ is $G_d(s)$-locally optimal and the following holds:*

$$N_{G_d(s)}(i) \supseteq N_{G_c(s)}(\geq i), \quad \forall i \in \mathcal{N}. \quad (11)$$

*Then $\pi$ is globally optimal.*

*Remark* 4.5. We write $G_d(s)$ and $G_c(s)$ to emphasize that the theorem can apply to problems where the CG may vary across states. For brevity, unless otherwise specified, we assume a fixed CG across

all states and denote it by $G_c$, with the corresponding fixed ADG denoted by $G_d$. Nevertheless, all subsequent results can immediately extend directly to state-dependent CG settings.

This theorem indicates that the ADG can be designed from the CG to guarantee that $G_d$-local optimality implies global optimality. Two special cases illustrate this principle: (i) When both $G_c$ and $G_d$ are empty, condition (11) reduces to $N_d(i) = N_c(\geq i) = \varnothing$, in which case the $Q$-function admits a VDN decomposition and any Nash equilibrium is globally optimal, consistent with Dou et al. (2022). (ii) When $G_c$ is complete and $G_d$ is fully dense, condition (11) is also satisfied. Thus, for any CG, a fully dense ADG guarantees global optimality, since every CG is a subgraph of the complete graph.

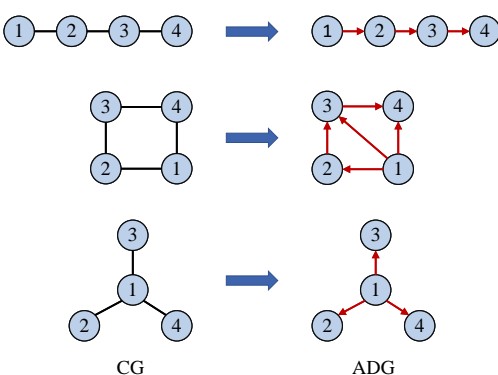

Figure 4: ADGs generated by Algorithm 2 for CG topologies: line, ring, and star.

If the agent indices are predetermined, replacing the superset relationship with an equality in condition (11) uniquely yields the sparsest ADG. However, the choice of index order strongly influences the sparsity of $G_d$, as shown in Figure 3. Determining the optimal index order is analogous to finding the optimal elimination order in variable elimination (VE), an NP-complete problem (Kok & Vlassis, 2006). Despite this complexity, practical heuristics such as the greedy algorithm described in Appendix E (Algorithm 2) can be employed. Figure 4 illustrates the resulting ADGs for several simple CG topologies.

# 5 CONVERGENCE OF ACTION-DEPENDENT POLICY

## 5.1 CONVERGENCE TO $G_d$-LOCALLY OPTIMAL POLICY

In this section, we introduce a policy iteration algorithm for MARL in the tabular setting. This algorithm highlights the advantage of employing action-dependent policies, enabling convergence to a $G_d$-locally optimal policy rather than merely an agent-by-agent optimal one.

Our approach extends the multi-agent policy iteration (MPI) framework proposed in Bertsekas (2021), which decomposes the joint policy update step of standard policy iteration (PI) (Sutton, 2018) into sequential updates of individual agents' policies, thereby mitigating the computational complexity of PI. However, MPI guarantees convergence only to an agent-by-agent optimal policy, which is often suboptimal. To address this limitation, we propose Algorithm 1, which incorporates action-dependent policies into the MPI framework and ensures convergence to a $G_d$-locally optimal policy.

---

**Algorithm 1** Action-Dependent Multi-Agent Policy Iteration

Initialize policies $\pi_i^1$, $i \in \mathcal{N}$, with ADG $G_d$ under every $s \in \mathcal{S}$
**for** $k = 1, 2, \dots$ **do**
    // *Policy Evaluation*
    Compute $V^{\pi^k}$ by solving $V = \mathcal{T}_{\pi^k} V$ and derive $Q^{\pi^k}$ from $V^{\pi^k}$
    // *Policy Improvement*
    **for** $i = 1, 2, \dots, n$ **do**
        Update $\pi_i^{k+1}$ for every $(s, a_{<i})$ pair by

$$\pi_i^{k+1}(s, a_{<i}) \leftarrow \arg\max_{a_i} Q^{\pi^k}(s, \pi_{(<i), N_d(i)}^{k+1}(s, a_{N_d(i)}), a_i, \pi_{(>i), N_d[i]}^k(s, a_{N_d[i]})). \quad (12)$$

    **end for**
**end for**

---

Since the policies of agents in $(< i)$ are always updated before agent $i$, the update rule (12) can always be applied in succession. Since the image set of $\arg\max$ may have multiple values, we arbitrarily select one of them. Specifically, if $\pi_i^k(s, a_{<i})$ is already in the image set, then we prioritize selecting $\pi_i^k(s, a_{<i})$. Note that while the update is specified for every $(s, a_{<i})$ pair, the $\arg\max$ in (12) only depends on $(s, a_{N_d(i)})$, so the actual computation only needs to be performed for every $(s, a_{N_d(i)})$ pair. When update rule no longer changes $\pi_i^k$ for any $(s, a_{<i})$, the algorithm reaches convergence. The following theorem describes the convergence property:

**Theorem 5.1** (Convergence of Algorithm 1, proof in Appendix D). *Let $\{\pi^k\}_{k=1}^{\infty}$ be the policy sequence generated by Algorithm 1. Then $\{\pi^k\}_{k=1}^{\infty}$ converges to a $G_d$-locally optimal policy in a finite number of steps.*

It is straightforward to verify that once all individual policies converge, the joint policy is $G_d$-locally optimal. Thus, the main challenge in proving this theorem lies in establishing convergence of all individual policies. Chen & Zhang (2023) studied action-dependent policies in the policy gradient method and encountered a similar issue, which they bypassed by assuming that individual policies always converge. In contrast, we show that the joint policy converges regardless of whether individual policies converge directly (see Appendix Lemma D.1). Although convergence of the joint policy does not automatically imply $G_d$-local optimality, we can inductively establish that all individual policies converge from the convergence of the joint policy (see Appendix Lemma D.6). Therefore, our policy iteration method does not require additional assumptions and provides a complete resolution to this challenge.

## 5.2 CONVERGENCE TO GLOBALLY OPTIMAL POLICY

When the CG of an MDP is fixed, independent of both the state and the joint policy (e.g., polymatrix games), we can construct an ADG that satisfies (11) based on the CG, and then apply Algorithm 1 to update policies under this ADG. Upon convergence, Theorem 5.1 ensures that the resulting policy is $G_d$-locally optimal, and by Theorem 4.4, it is also globally optimal.

**Corollary 5.2.** *Assume $G_c$ is the CG of the state-action value function $Q^\pi$ for all policies $\pi$ and states $s$. Let $\{\pi^k\}_{k=1}^{\infty}$ be the policy sequence generated by Algorithm 1 with an ADG $G_d$. If $G_c$ and $G_d$ satisfy (11), then $\{\pi^k\}_{k=1}^{\infty}$ converges to a globally optimal policy in a finite number of steps.*

In more general scenarios, the CG may be dynamic. To guarantee convergence to an optimal solution, the ADG must also evolve accordingly. Such changes in the CG are typically driven by both the state and the joint policy. We denote this dependence by $G_c(s, \pi)$. For state-dependent changes, it suffices to design a distinct ADG for each state, ensuring that Equation (11) remains valid. In contrast, for policy-dependent changes, the ADG should be updated after each policy iteration to preserve convergence to the globally optimal policy. A key challenge arises here: directly modifying the ADG may alter the structural dependencies of individual policies. For instance, under one ADG, agent $i$ may depend only on the action of agent $j$, whereas under another ADG it may additionally depend on the actions of both $j$ and $k$. If the policy of agent $i$ lacks the interface to accommodate these additional dependencies, direct modification of the ADG becomes infeasible.

To address this issue, we could employ an indirect construction. Specifically, we build a set of individual policies such that, although their functional forms may differ before and after the ADG update, their induced joint policies remain identical. For deterministic policies, this construction is straightforward. Given a joint policy $\tilde{\pi}$ with independent components $\tilde{\pi}(s) = (\tilde{\pi}_1(s), \ldots, \tilde{\pi}_n(s))$, we define a set of action-dependent policies $\{\pi_i\}_{i \in \mathcal{N}}$ with new ADG such that,

$$\pi_1(s) \leftarrow \tilde{\pi}_1(s), \quad \pi_2(s, \pi_1(s)) \leftarrow \tilde{\pi}_2(s), \quad \ldots, \quad \pi_n(s, \tilde{\pi}_1(s), \ldots, \tilde{\pi}_{n-1}(s)) \leftarrow \tilde{\pi}_n(s). \quad (13)$$

This guarantees that the joint policies before and after the ADG update are equivalent. Consequently, if the CG changes after (12), we reconstruct the new policy $\pi_i^{k+1}$ according to (13). This ensures that the ADG and CG continue to satisfy Equation (11), thereby allowing Algorithm 1 to achieve the optimal solution.

**Corollary 5.3** (Proof in Appendix D). *Assume $G_c(s, \pi)$ is the CG of the state-action value $Q^\pi$ for joint policy $\pi$ and state $s$. Let $\{\pi^k\}_{k=1}^{\infty}$ be the policy sequence generated by Algorithm 1, with policy reconstruction according to (13) upon CG change. Then $\{\pi^k\}_{k=1}^{\infty}$ converges to a globally optimal policy in a finite number of steps.*

# 6 EXPERIMENTS

## 6.1 COORDINATION POLYMATRIX GAMES

To validate our theoretical results, we evaluate Algorithm 1 on polymatrix games with CGs of different topologies: star (5 agents), ring (5 agents), tree (7 agents), and mesh (9 agents). The payoff matrices are randomly generated, with the maximum reward set equal to the number of CG edges. We compare three ADG types: sparse (generated by Algorithm 2 to satisfy (11)), fully dense, and empty. Additional experimental details are provided in Appendix F. Figure 5 reports the learning curves averaged over 100 runs. Both sparse and dense ADGs consistently reach the reward upper bound, with their learning curves being very close. In contrast, empty ADGs often become trapped in suboptimal Nash equilibria.

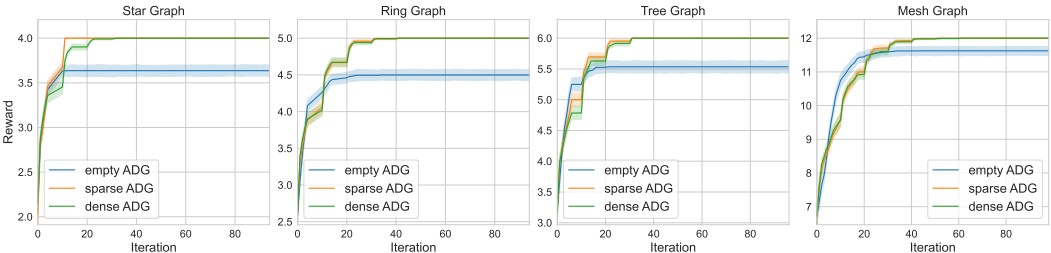

Figure 5: Results of coordination polymatrix game.

To assess computational efficiency, Figure 6 (right) shows the average per-iteration runtime across 100 experiments when each agent has two actions. Shaded areas represent the 95% confidence intervals computed over multiple independent runs (similarly hereafter). As runtime is primarily determined by policy dimension and dimensions of dense ADG policies are independent of CG structure, their curves nearly overlap. Due to exponential growth in policy dimension, dense ADGs incur substantially higher costs as agent number increases, whereas sparse ADGs maintain scalability.

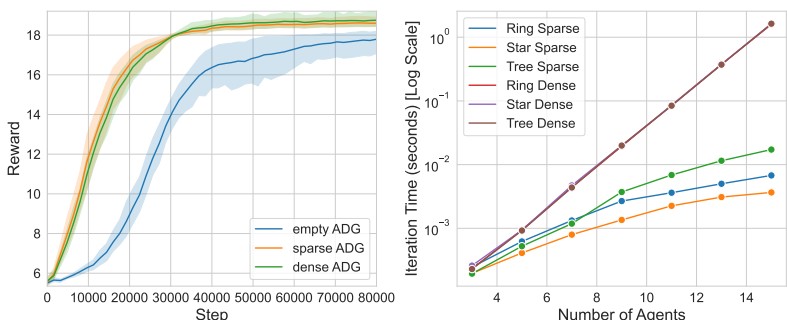

Figure 6: Results of MAPPO on star CG (left) and average time per iteration (right).

We also extend the MAPPO algorithm to incorporate action-dependent policies to examine the convergence of action-dependent policies under more practical learning settings. Nevertheless, ADGs are not tailored to any specific MARL algorithm. Any method that includes an actor module can be adapted to the action-dependent policy. For algorithms based on independent actor, one can simply replace the independent actor with an action-dependent actor following the ADG structure. For auto-regressive actor, the adaptation amounts to removing action information that lies outside the ADG. For MAPPO, we transform the independent policy $\pi_{\theta_i}(a_i|s)$ into the action-dependent form $\pi_{\theta_i}(a_i|s, a_{N_d(i)})$. This requires a corresponding modification of the optimization objective to properly handle the action-dependent policy. Consider MAPPO (Yu et al., 2022), where the original objective is

$$\mathcal{L}(\theta) = \mathbb{E}_{s\sim\mathcal{D}, a\sim\pi_{\theta_{\text{old}}}} \left[ \sum_{i=1}^{n} \min\left( r_{\theta_i}(a_i, s) A_{\pi_{\theta_{\text{old}}}}(s, a), \ \text{clip}(r_{\theta_i}(a_i, s), 1 \pm \varepsilon) A_{\pi_{\theta_{\text{old}}}}(s, a) \right) \right],$$

where $r_{\theta_i}(a_i, s) = \frac{\pi_{\theta_i}(a_i|s)}{\pi_{\theta_{i,\text{old}}}(a_i|s)}$, and $\mathcal{D}$ denotes the distribution in the replay buffer. To adapt this objective for action-dependent policies, we replace $r_{\theta_i}(a_i, s)$ with $r_{\theta_i}(a_i, s, a_{N_d(i)}) = \frac{\pi_{\theta_i}(a_i|s, a_{N_d(i)})}{\pi_{\theta_{i,\text{old}}}(a_i|s, a_{N_d(i)})}$.

Figure 6 (left) presents results of the extended MAPPO on polymatrix games with a star CG, using fixed payoff matrices (maximum reward 20) and 10 random seeds. As the value of the largest suboptimal point is 18, both sparse and dense ADGs successfully escape all suboptimal points. The slight deviations from the maximum reward are primarily attributed to exploratory behavior caused by entropy regularization. These results demonstrate that even in non-tabular settings parameterized by neural networks, Theorem 4.4 provides a reliable design principle for sparse ADGs. In future work, we also plan to provide a rigorous theoretical analysis to substantiate these empirical findings.

## 6.2 ATSC

To further examine practical applicability, we evaluate the extended MAPPO on adaptive traffic signal control (ATSC) problem, a benchmark with a natural coordination structure. Experiments are conducted on 2x2 (4 agents), 3x3 (9 agents), and 4x4 (16 agents) road networks using the Simulation of Urban Mobility (SUMO) platform Lopez et al. (2018) and SUMO-RL Alegre (2019). The CG is defined by adjacency between intersections, and sparse ADGs are derived with Algorithm 2. To verify the computational efficiency improvement of sparse ADGs in deep learning settings, we measure the FLOPs required for a single forward pass of the action-dependent policy employed in the ATSC, as presented in Table 1. Detailed experimental protocols and hyperparameters are given in Appendix F.



Figure 7: 3x3 road network.

As shown in Figure 8, sparse ADGs achieve performance comparable to dense ADGs, and both outperform empty ADGs. This indicates that even with approximate CG structures, sparse ADGs retain the efficiency of action-dependent policies without sacrificing optimality.

Table 1: FLOPs of a single forward pass in ATSC.

| environment | dense ADG | sparse ADG |
|---|---|---|
| 2x2grid | 45312 | 42496 |
| 3x3grid | 140112 | 104832 |
| 4x4grid | 412672 | 232448 |

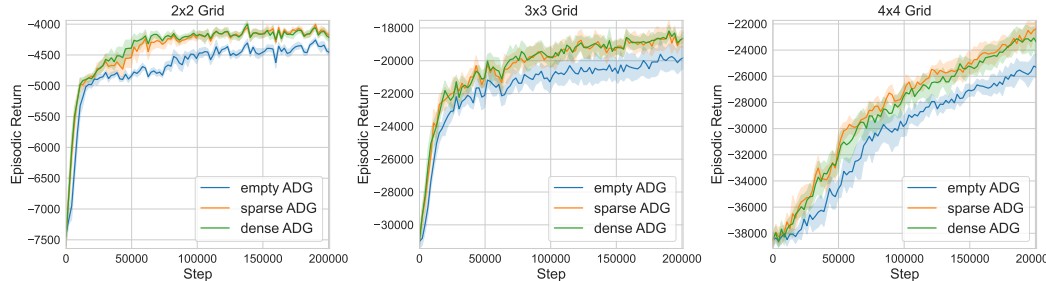

Figure 8: Results of ATSC.

## 7 CONCLUSION

In this work, we established a theoretical framework for action-dependent policies in multi-agent reinforcement learning by introducing the ADG and the notion of $G_d$-locally optimal policies. We further identified conditions under which these policies coincide with globally optimal solutions in coordination graph structured problems, and proposed a policy iteration algorithm with guaranteed convergence. Finally, we validated our theory and demonstrated its practical potential through experiments on polymatrix games and adaptive traffic signal control. Recognizing that complex environments may involve unknown CGs, hypergraph CGs, we aim to explore the adaptability and potential of ADGs in these challenging settings in future research.

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

## APPENDIX

## A  THE USE OF LARGE LANGUAGE MODELS

We used LLMs for improving the grammar and clarity of the manuscript. All scientific content, ideas, and analysis are original and authored by the listed contributors.

## B  MATHEMATICAL PRELIMINARIES

In the appendix, we first reformulate the MAMDP as a sequentially expanded MDP (SEMDP) (Ye et al., 2022; Li et al., 2023; Bhattacharya & Bal, 2025), and then establish our proofs on this reformulation. The SEMDP transforms a multi-agent system into a single-agent MDP by expanding the state space. Importantly, the SEMDP reformulation is non-essential: it does not introduce any new assumptions, and our results can still be proven directly under the MAMDP formulation. However, introducing the SEMDP greatly simplifies the notation used in the proofs.

Given an MAMDP $\langle \mathcal{N}, \mathcal{S}, \mathcal{A}, \tilde{P}, \tilde{r}, \gamma^n \rangle$, we construct a SEMDP denoted by $\langle \mathcal{N}, \mathcal{S}, \mathcal{A}, P, r, \gamma \rangle$, where

- $\mathcal{S}$ is identical to the state space in the MAMDP;
- $\mathcal{A} = \prod_{i=1}^{n} \mathcal{A}_i$ is identical to the joint action space in the MAMDP;
- $\mathcal{Z} = \bigcup_{i \in \mathcal{N}} \mathcal{Z}_i$ is the expanded state space, where $\mathcal{Z}_i = \mathcal{S} \times \prod_{j=1}^{i-1} \mathcal{A}_j$ is the individual expanded state space for agent $i$;
- $P : \mathcal{Z} \times \mathcal{Z} \times \bigcup_{i \in \mathcal{N}} \mathcal{A}_i \to [0, 1]$ is the transition kernel of the SEMDP;
- $r : \bigcup_{i \in \mathcal{N}} (\mathcal{Z}_i \times \mathcal{A}_i) \to \mathbb{R}$ is the reward function of the SEMDP;

The SEMDP transition kernel is defined as

$$P((s, a_{\leq i})|(s, a_{<i}), a_i) = 1, \quad \forall s \in \mathcal{S}, a_{\leq i} \in \mathcal{A}_{\leq i}, 0 < i < n, \tag{14}$$

$$P(s'|(s, a_{<n}), a_n) = \tilde{P}(s'|s, a), \quad \forall s \in \mathcal{S}, a_{\leq n} \in \mathcal{A}_{\leq n}. \tag{15}$$

The reward function is defined as

$$r((s, a_{<i}); a_i) = 0, \quad \forall s \in \mathcal{S}, a_{\leq i} \in \mathcal{A}_{\leq i}, 0 < i < n, \tag{16}$$

$$r((s, a_{<n}); a_n) = \tilde{r}(s, a), \quad \forall s \in \mathcal{S}, a \in \mathcal{A}. \tag{17}$$

In the appendix, the joint policy in the SEMDP is denoted by $\pi : \mathcal{Z} \to \bigcup_{i \in \mathcal{N}} \mathcal{A}$, where

$$\pi(s, a_{<i}) \in \mathcal{A}_i, \quad \forall s \in \mathcal{S}, a_{<i} \in \mathcal{A}_{<i}.$$

The individual policy of agent $i$ is defined as $\pi_i := \pi|_{\mathcal{Z}_i}$, which restricts the function to $\mathcal{Z}_i$. This formulation is identical to the individual policy in the MAMDP. To distinguish between the joint policies in MAMDP and SEMDP, we rewrite the joint policy in the MAMDP as $\hat{\pi} := \pi_{\mathcal{N}, \varnothing}^k = (\pi_{1,\varnothing}, \ldots, \pi_{n,\varnothing})$.

In the SEMDP, the state value function takes the form $V : \mathcal{Z} \to \mathbb{R}$, and the state-action value function takes the form $Q : \bigcup_{i \in \mathcal{N}} (\mathcal{Z}_i \times \mathcal{A}_i) \to \mathbb{R}$. For any $V \in \mathcal{R}(\mathcal{Z})$, we define

$$Q^V(s, a_{<i}; a_i) := r(s, a_{<i}; a_i) + \gamma \sum_{z' \in \mathcal{Z}} P(z' \mid (s, a_{<i}), a_i) V(z'). \tag{18}$$

The Bellman operator $\mathcal{T}_\pi$ and the optimal Bellman operator $\mathcal{T}$ are given by

$$\mathcal{T}_\pi V(z) = Q^V(s, a_{<i}; \pi_i(s, a_{<i})), \quad \mathcal{T}V(z) = \max_{a_i} Q^V(s, a_{<i}; a_i). \tag{19}$$

The state value function $V^\pi$ is the fixed point of $\mathcal{T}_\pi$, and the state-action value function is $Q^\pi := Q^{V^\pi}$. The optimal state value function $V^*$ is the fixed point of $\mathcal{T}$, and $Q^* := Q^{V^*}$. The semicolon in $Q$ and $r$ highlights the action dependence. However, $Q(s, a_{<n}; a_n)$ can also be regarded as the value function of the full joint action and thus is sometimes written simply as $Q(s, a)$.

**Proposition B.1.** *We summarize several fundamental properties of the SEMDP:*

1. *$V^{\hat{\pi}}(s) = \gamma^{1-n} V^\pi(s)$, $Q^{\hat{\pi}}(s, a) = \gamma^{1-n} Q^\pi(s, a)$, where $V^{\hat{\pi}}$ and $Q^{\hat{\pi}}$ denotes the state and state-action value function in the original MAMDP.*

2. *$Q^\pi(s, a_{<i}; a_i) = \gamma V^\pi(s, a_{\leq i}), \quad 1 \leq i < n.$*

3. *$Q^\pi(s, a_{<i}; a_i) = \gamma^{n-i} Q^\pi(s, \pi_{(<n),(\leq i)}(s, a_{\leq i}); \pi_{n,(\leq i)}(s, a_{\leq i})), \quad 1 \leq i < n.$*

*Proof.* (i) We expand the definition of $V^{\hat{\pi}}$ in the MAMDP:

$$V^{\hat{\pi}}(s) = \mathbb{E}\left[\sum_{t=0}^{\infty} \gamma^{nt} r(s_t, \hat{\pi}(s_t)) \bigg| s_{t+1} \sim P(\cdot|s_t, \hat{\pi}(s_t)), s_0 = s\right]$$

$$= \mathbb{E}\left[\sum_{t=0}^{\infty} \sum_{i=1}^{n} \gamma^{nt+i-n} r(s_t, \pi_{<i,\varnothing}(s_t); \pi_{i,\varnothing}(s_t)) \bigg| s_{t+1} \sim P(\cdot|s_t, \hat{\pi}(s_t)), s_0 = s\right]$$

$$= \mathbb{E}\left[\gamma^{1-n} \sum_{t=0}^{\infty} \sum_{i=1}^{n} \gamma^{nt+i-1} r(s_t, \pi_{<i,\varnothing}(s_t); \pi_{i,\varnothing}(s_t)) \bigg| s_{t+1} \sim P(\cdot|s_t, \hat{\pi}(s_t)), s_0 = s\right]$$

$$= \gamma^{1-n} \mathbb{E}\left[\sum_{t'=0}^{\infty} \gamma^{t'} r(z_{t'}; \pi(z_{t'})) \bigg| z_{t'+1} \sim P(\cdot|z_{t'}, \pi(z_{t'})), z_0 = s\right]$$

$$= \gamma^{1-n} V^\pi(s)$$

Similarly, we can derive that $Q^{\hat{\pi}}(s) = \gamma^{1-n} Q^\pi(s)$.

(ii) For $1 \leq i < n$, we compute:

$$Q^\pi(s, a_{<i}; a_i) = r(s, a_{<i}; a_i) + \gamma \sum_{z' \in \mathcal{Z}} P(z' \mid (s, a_{<i}), a_i) V^\pi(z')$$

$$= \gamma V^\pi(s, a_{\leq i}).$$

(iii) Repeatedly unrolling the recursion yields:

$$Q^\pi(s, a_{<i}; a_i) = \gamma V^\pi(s, a_{\leq i})$$
$$= \gamma Q^\pi(s, a_{\leq i}; \pi_{i+1,(\leq i)}(s, a_{\leq i}))$$
$$= \cdots = \gamma^{n-i} Q^\pi(s, \pi_{(<n),(\leq i)}(s, a_{\leq i}); \pi_{n,(\leq i)}(s, a_{\leq i})).$$
$\square$

From Proposition B.1(i), we see that the value functions in the SEMDP and the MAMDP are equivalent up to a constant factor. Therefore, it can be easily verified that (1) an optimal policy in the SEMDP remains optimal in the original MAMDP, (2) $Q^\pi$ and $Q^{\tilde{\pi}}$ have the same CG, and (3) in update rule (12), replacing the $Q$-function with its SEMDP counterpart does not affect the update outcome:

$$\pi_i^{k+1}(s, a_{<i}) \leftarrow \arg\max_{a_i} Q^{\pi^k}(s, \pi_{(<i),N_d(i)}^{k+1}(s, a_{N_d(i)}); a_i). \tag{20}$$

## C   PROOF OF OPTIMALITY

In the appendix, we generalize the concept of the dependency set to more general functions to simplify the description of subsequent proofs.

**Definition C.1** (Dependency Set). Let $f : \mathcal{A}_S \to \mathcal{Y}$ be a mapping defined on the joint action space of a subset of agents $S \subseteq \mathcal{N}$. A subset $C \subseteq S$ is called a dependency set of $f$ if for any $s \in \mathcal{S}$, $a_C \in \mathcal{A}_C$, $a'_{S \setminus C}, a_{S \setminus C} \in \mathcal{A}_{S \setminus C}$, the following holds:

$$f(a_C, a'_{S \setminus C}) = f(a_C, a_{S \setminus C}).$$

For notational convenience, we may permute the order of variables when writing a function, but the evaluation of the function always follows the ordering of variables according to their agent indices.

Since the form of the $Q$-function changes in the SEMDP setting, we restate the definition of $G_d$-locally optimal policies for SEMDPs. Note that, according to Proposition B.1 (i) and (iii), this definition is equivalent to the one given in the main text.

**Definition C.2** ($G_d$-locally Optimal in SEMDP). Let $G_d$ be a DAG. A joint policy $\pi$ is $G_d$-locally optimal under $s \in \mathcal{S}$ if, for any $a_{N_d(i)} \in \mathcal{A}_{N_d(i)}$, the following holds:

$$Q^\pi(s, \pi_{(<i),N_d(i)}(s, a_{N_d(i)}); \pi_{i,N_d(i)}(s, a_{N_d(i)})) = \max_{a_i} Q^\pi(s, \pi_{(<i),N_d(i)}(s, a_{N_d(i)}); a_i). \tag{21}$$

**Lemma C.3.** *For any fixed $s \in \mathcal{S}$, let $G_d$ be a DAG. Suppose a joint policy $\pi$ is $G_d$-locally optimal at $s$. If for every $i \in \mathcal{N}$, the set $S_i$ is the dependency set of the function $\arg\max_{a_i} Q^\pi(s, a_{(<i)}; a_i)$ with respect to $a_{(<i)}$ at $s$, and if $N_d(i) \supseteq S_i$, then $\pi$ is globally optimal.*

*Proof.* Fix any $s \in \mathcal{S}$. Since $\pi$ is $G_d$-locally optimal, we have

$$\pi_i(s, a_{<i}) \in \arg\max_{a_i} Q^\pi(s, \pi_{(<i),N_d(i)}(s, a_{N_d(i)}); a_i)$$
$$= \arg\max_{a_i} Q^\pi(s, a_{N_d(i)}, \pi_{(<i)\setminus N_d(i),N_d(i)}(s, a_{N_d(i)}); a_i). \tag{22}$$

Because $N_d(i) \supseteq S_i$, where $S_i$ is the dependency set of $\arg\max_{a_i} Q^\pi(s, a_{(<i)}; a_i)$, it follows that

$$\pi_i(s, a_{<i}) \in \arg\max_{a_i} Q^\pi(s, a_{<i}; a_i), \quad \forall a_{<i} \in \mathcal{A}_{<i}. \tag{23}$$

Therefore,

$$V^\pi(s, a_{<i}) = \mathcal{T}_\pi V^\pi(s, a_{<i}) = Q^\pi(s; a_{<i}; \pi_i(s, a_{<i}))$$
$$= \max_{a_i} Q^\pi(s; a_{<i}; a_i) = \mathcal{T} V^\pi(s, a_{<i}), \tag{24}$$

which shows that $V^\pi$ is a fixed point of $\mathcal{T}$. Hence, $\pi$ is globally optimal. $\square$

**Lemma C.4.** *Let $Q$ be a state–action value function, and let $G_c = (\mathcal{N}, \mathcal{E}_c)$ be the CG of $Q$ at state $s$. Then for any $i \in \mathcal{N}$, there exist functions $Q_1 : \mathcal{S} \times \mathcal{A}_{N_c[\geq i]} \to \mathbb{R}$ and $Q_2 : \mathcal{S} \times \mathcal{A}_{<i} \to \mathbb{R}$ such that*

$$Q(s, a) = Q_1(s, a_{N_c[\geq i]}) + Q_2(s, a_{<i}), \quad \forall s \in \mathcal{S}, a \in \mathcal{A}, \tag{25}$$

*where $N_c[\geq i] := N_c(\geq i) \cup (\geq i)$.*

*Proof.* Decomposing $Q$ according to the structure of $G_c$ yields

$$Q(s, a) = \left( \sum_{(j,k) \in \mathcal{E}_c[\geq i, <i]} + \sum_{(j,k) \in \mathcal{E}_c[\geq i]} + \sum_{(j,k) \in \mathcal{E}_c[<i]} \right) Q_{jk}(s, a_j, a_k), \tag{26}$$

where $\mathcal{E}_c[\geq i, < i]$ denotes the subset of $\mathcal{E}_c$ containing edges between vertices in the sets $\geq i$ and $< i$, and $\mathcal{E}_c[\geq i] := \mathcal{E}_c[\geq i, \geq i]$ containing edges within the set $\geq i$. Noting that $\mathcal{E}_c[\geq i, < i] = \mathcal{E}_c[\geq i, N_c(\geq i)]$, thus, we can rewrite:

$$Q(s, a) = \left( \sum_{(j,k) \in \mathcal{E}_c[\geq i, N_c(\geq i)]} + \sum_{(j,k) \in \mathcal{E}_c[\geq i]} + \sum_{(j,k) \in \mathcal{E}_c[N_c(\geq i)]} + \sum_{(j,k) \in \mathcal{E}_c[<i] \setminus \mathcal{E}_c[N_c(\geq i)]} \right) Q_{jk}(s, a_j, a_k). \tag{27}$$

Define

$$Q_1(s, a_{N_c[\geq i]}) := \left( \sum_{(j,k) \in \mathcal{E}_c[\geq i, N_c(\geq i)]} + \sum_{(j,k) \in \mathcal{E}_c[\geq i]} + \sum_{(j,k) \in \mathcal{E}_c[N_c(\geq i)]} \right) Q_{jk}(s, a_j, a_k), \tag{28}$$

and

$$Q_2(s, a_{<i}) := \sum_{(j,k) \in \mathcal{E}_c[<i] \setminus \mathcal{E}_c[N_c(\geq i)]} Q_{jk}(s, a_j, a_k). \tag{29}$$

Then (25) follows. $\qquad\square$

**Proof of Theorem 4.4**

*Proof.* For any fixed state $s$, we abbreviate $N_c(i) = N_{G_c(s)}(i)$ and $N_d(i) = N_{G_d(s)}(i)$. We proceed by induction.

**Base case:** We aim to show that

$$\max_{a_n} Q^\pi(s, a_{<n}; a_n) = Q^\pi(s, a_{<n}; \pi_n(s, a_{<n})), \tag{30}$$

and that $N_c(\geq n)$ is the dependency set of $\arg\max_{a_n} Q^\pi(s, a_{<n}; a_n)$.

By Lemma C.4, there exist functions $Q_1$ and $Q_2$ such that

$$Q^\pi(s, a_{<n}; a_n) = Q_1(s, a_{N_c[n]}) + Q_2(s, a_{<n}). \tag{31}$$

Consider

$$Q^\pi(s, a_{(<n) \setminus N_c(n)}, a_{N_c(n)}; a_n) - Q^\pi(s, a'_{(<n) \setminus N_c(n)}, a_{N_c(n)}; a_n)$$
$$= Q_2(s, a_{(<n) \setminus N_c(n)}, a_{N_c(n)}) - Q_2(s, a'_{(<n) \setminus N_c(n)}, a_{N_c(n)}), \tag{32}$$

which is independent of $a_n$. Thus, the maximizing $a_n$ is unaffected by $a_{<n \setminus N_c(n)}$, implying that

$$\arg\max_{a_n} Q^\pi(s, a_{<n \setminus N_c(n)}, a_{N_c(n)}; a_n) = \arg\max_{a_n} Q^\pi(s, a'_{<n \setminus N_c(n)}, a_{N_c(n)}; a_n). \tag{33}$$

Therefore, $N_c(n) = N_c(\geq n)$ forms the dependency set of $\arg\max_{a_n} Q^\pi(s, a_{<n}; a_n)$. Since $\pi$ is $G_d$-locally optimal and $N_d(n) \supseteq N_c(\geq n)$, it follows that

$$\pi_n(s, a_{<n}) \in \arg\max_{a_n} Q^\pi(s, a_{<n}; a_n), \tag{34}$$

and hence

$$\max_{a_n} Q^\pi(s, a_{<n}; a_n) = Q^\pi(s, a_{<n}; \pi_n(s, a_{<n})). \tag{35}$$

**Induction step:** Assume for some $i + 1$ that

$$\max_{a_{\geq i+1}} Q^\pi(s, a_{<n}; a_n) = Q^\pi(s, \pi_{(<n),(<i+1)}(s, a_{<i+1}); \pi_{n,(<i+1)}(s, a_{<i+1})), \quad (36)$$

and that $N_c(\geq i + 1)$ is the dependency set of $\arg\max_{a_{i+1}} Q^\pi(s, a_{<(i+1)}; a_{i+1})$. We now prove the case for $i$.

By Lemma C.4 and Proposition B.1 (iii), there exist functions $Q_1$ and $Q_2$ such that

$$\begin{aligned}
Q^\pi(s, a_{<i}; a_i) &= \gamma^{n-i} Q^\pi(s, \pi_{(<n),(\leq i)}(s, a_{\leq i}); \pi_{n,(\leq i)}(s, a_{\leq i})) \\
&= Q_1(s, \pi_{N_c[\geq i],(\leq i)}(s, a_{\leq i})) + Q_2(s, \pi_{(<i),(<i)}(s, a_{<i})).
\end{aligned} \quad (37)$$

From Equation (36), we obtain

$$\pi_{(\geq i+1),(\leq i)}(s, a_{\leq i}) \in \arg\max_{a_{\geq i+1}} Q_1(s, a_{N_c[\geq i]}), \quad (38)$$

and hence

$$\begin{aligned}
&Q_1(s, a_i, a_{N_c(\geq i)}, \pi_{(\geq i+1),(\leq i)}(s, a_{N_c(\geq i)}, a_{(<i)\setminus N_c(\geq i)})) \\
&= \max_{a_{\geq i+1}} Q_1(s, a_{N_c[\geq i]}) \\
&= Q_1(s, a_i, a_{N_c(\geq i)}, \pi_{(\geq i+1),(\leq i)}(s, a_{N_c(\geq i)}, a'_{(<i)\setminus N_c(\geq i)})).
\end{aligned} \quad (39)$$

Following the same argument as in (32),

$$\begin{aligned}
&Q^\pi(s, a_{N_c(\geq i)}, a_{(<i)\setminus N_c(\geq i)}; a_i) - Q^\pi(s, a_{N_c(\geq i)}, a'_{(<i)\setminus N_c(\geq i)}; a_i) \\
=&Q_1(s, \pi_{N_c(\geq i),(<i)}(s, a_{N_c(\geq i)}, a_{(<i)\setminus N_c(\geq i)}), a_i) + Q_2(s, \pi_{(<i),(<i)}(s, a_{N_c(\geq i)}, a_{(<i)\setminus N_c(\geq i)})) \\
&- Q_1(s, \pi_{N_c(\geq i),(<i)}(s, a_{N_c(\geq i)}, a'_{(<i)\setminus N_c(\geq i)}), a_i) - Q_2(s, \pi_{(<i),(<i)}(s, a_{N_c(\geq i)}, a'_{(<i)\setminus N_c(\geq i)})) \\
=&Q_2(s, \pi_{(<i),(<i)}(s, a_{N_c(\geq i)}, a_{(<i)\setminus N_c(\geq i)})) - Q_2(s, \pi_{(<i),(<i)}(s, a_{N_c(\geq i)}, a'_{(<i)\setminus N_c(\geq i)})).
\end{aligned} \quad (40)$$

Therefore,

$$\arg\max_{a_i} Q^\pi(s, a_{N_c(\geq i)}, a_{(<i)\setminus N_c(\geq i)}; a_i) = \arg\max_{a_i} Q^\pi(s, a_{N_c(\geq i)}, a'_{(<i)\setminus N_c(\geq i)}; a_i), \quad (41)$$

which implies that $N_c(\geq i)$ is the dependency set of $\arg\max_{a_i} Q^\pi(s, a_{<i}; a_i)$. Moreover, since $\pi$ is $G_d$-locally optimal and $N_d(i) \supseteq N_c(\geq i)$, it follows that

$$\begin{aligned}
\pi_i(s, a_{<i}) &\in \arg\max_{a_i} Q^\pi(s, \pi_{(<i),N_d(i)}(s, a_{N_d(i)}); a_i) \\
&= \arg\max_{a_i} Q^\pi(s, a_{N_c(\geq i)}, \pi_{(<i)\setminus N_c(\geq i),N_d(i)}(s, a_{N_d(i)}); a_i) \\
&= \arg\max_{a_i} Q^\pi(s, a_{<i}; a_i).
\end{aligned} \quad (42)$$

Consequently, by Proposition B.1 (iii), it holds that

$$\begin{aligned}
&Q^\pi(s, \pi_{(<n),(<i)}(s, a_{<i}); \pi_{n,(<i)}(s, a_{<i})) \\
=&\gamma^{i-n} Q^\pi(s, a_{<i}; \pi_i(s, a_{<i})) \\
=&\max_{a_i} \gamma^{i-n} Q^\pi(s, a_{<i}; a_i) \\
=&\max_{a_i} Q^\pi(s, \pi_{(<n),(<i+1)}(s, a_{<i+1}); \pi_{n,(<i+1)}(s, a_{<i+1})) \\
=&\max_{a_i} \max_{a_{\geq i+1}} Q^\pi(s, a_{<n}; a_n) = \max_{a_{\geq i}} Q^\pi(s, a_{<n}; a_n).
\end{aligned} \quad (43)$$

**Conclusion:** By induction, for any given $s$, $N_c(\geq i)$ is the dependency set of $\arg\max_{a_i} Q^\pi(s, a_{<i}; a_i)$ for all $i \in \mathcal{N}$. Together with the condition $N_d(i) \supseteq N_c(\geq i)$, Lemma C.3 guarantees that $\pi$ is globally optimal. $\qquad\square$

## D  PROOF OF CONVERGENCE

We first prove that the joint policy $\pi_{\mathcal{N},\varnothing}^k(s)$ in MAMDP converges.

**Lemma D.1.** *Let $\{\pi^k\}_{k=1}^\infty$ be the policy sequence generated by Algorithm 1. Then both $V^{\pi^k}(s)$ and $\pi_{\mathcal{N},\varnothing}^k(s)$ converge within a finite number of steps.*

*Proof.* From the update rule (20), for any $s \in \mathcal{S}, 0 \leq i \leq n$, we have

$$
\begin{aligned}
\mathcal{T}_{\pi^{k+1}} V^{\pi^k}(s, \pi_{<i,\varnothing}^{k+1}(s)) &= Q^{\pi^k}(s, \pi_{<i,\varnothing}^{k+1}(s); \pi_i^{k+1}(s, \pi_{<i,\varnothing}^{k+1}(s))) \\
&\geq Q^{\pi^k}(s, \pi_{<i,\varnothing}^{k+1}(s); \pi_i^k(s, \pi_{<i,\varnothing}^{k+1}(s))) \\
&= V^{\pi^k}(s, \pi_{<i,\varnothing}^{k+1}(s)).
\end{aligned}
\tag{44}
$$

We now proceed by induction. Assume that

$$
\mathcal{T}_{\pi^{k+1}}^j V^{\pi^k}(s, \pi_{<i,\varnothing}^{k+1}(s)) \geq \mathcal{T}_{\pi^{k+1}}^{j-1} V^{\pi^k}(s, \pi_{<i,\varnothing}^{k+1}(s)), \quad \forall s \in \mathcal{S}, \, 0 \leq i \leq n.
\tag{45}
$$

We now prove that

$$
\mathcal{T}_{\pi^{k+1}}^{j+1} V^{\pi^k}(s, \pi_{<i,\varnothing}^{k+1}(s)) \geq \mathcal{T}_{\pi^{k+1}}^j V^{\pi^k}(s, \pi_{<i,\varnothing}^{k+1}(s)), \quad \forall s \in \mathcal{S}, \, 0 \leq i \leq n.
\tag{46}
$$

When $i < n$,

$$
\begin{aligned}
\mathcal{T}_{\pi^{k+1}}^{j+1} V^{\pi^k}(s, \pi_{<i,\varnothing}^{k+1}(s)) &= \gamma \mathcal{T}_{\pi^{k+1}}^j V^{\pi^k}(s, \pi_{<i+1,\varnothing}^{k+1}(s)) \\
&\geq \gamma \mathcal{T}_{\pi^{k+1}}^{j-1} V^{\pi^k}(s, \pi_{<i+1,\varnothing}^{k+1}(s)) \\
&= \mathcal{T}_{\pi^{k+1}}^j V^{\pi^k}(s, \pi_{<i,\varnothing}^{k+1}(s)).
\end{aligned}
\tag{47}
$$

When $i = n$,

$$
\begin{aligned}
\mathcal{T}_{\pi^{k+1}}^{j+1} V^{\pi^k}(s, \pi_{<n,\varnothing}^{k+1}(s)) &= r(s, \pi_{\leq n,\varnothing}^{k+1}(s)) + \gamma \sum_{s'} P(s'|s, \pi_{\leq n,\varnothing}^{k+1}(s)) \mathcal{T}_{\pi^{k+1}}^i V^{\pi^k}(s') \\
&\geq r(s, \pi_{\leq n,\varnothing}^{k+1}(s)) + \gamma \sum_{s'} P(s'|s, \pi_{\leq n,\varnothing}^{k+1}(s)) \mathcal{T}_{\pi^{k+1}}^{i-1} V^{\pi^k}(s') \\
&= \mathcal{T}_{\pi^{k+1}}^j V^{\pi^k}(s, \pi_{<n,\varnothing}^{k+1}(s)).
\end{aligned}
\tag{48}
$$

Thus,

$$
V^{\pi^{k+1}}(s, \pi_{<i,\varnothing}^{k+1}(s)) = \lim_{j \to \infty} \mathcal{T}_{\pi^{k+1}}^j V^{\pi^k}(s, \pi_{<i,\varnothing}^{k+1}(s)) \geq V^{\pi^k}(s, \pi_{<i,\varnothing}^{k+1}(s)) \geq V^{\pi^k}(s, \pi_{<i,\varnothing}^k(s)).
\tag{49}
$$

By the monotone convergence theorem, $V^{\pi^k}(s, \pi_{<i,\varnothing}^k(s))$ converges. Since the policy space is finite, $V^{\pi^k}(s, \pi_{<i,\varnothing}^k(s)), \forall i \in \mathcal{N}, s \in \mathcal{S}$ converges within a finite number of steps.

Next, we prove by contradiction that $\pi_{\mathcal{N},\varnothing}^k(s)$ also converges. Suppose that for some $M$, $V^{\pi^k}(s)$ has already converged when $k \geq M$. Assume that for $k \geq M$, $\pi_{\mathcal{N},\varnothing}^k(s) \neq \pi_{\mathcal{N},\varnothing}^{k+1}(s)$. Let $i$ be the first

index such that $\pi_{i,\varnothing}^k(s) \neq \pi_{i,\varnothing}^{k+1}(s)$ while $\pi_{<i,\varnothing}^k(s) = \pi_{<i,\varnothing}^{k+1}(s)$. Then

$$
\begin{aligned}
V^{\pi^{k+1}}(s) &= \gamma V^{\pi^{k+1}}(s, \pi_{<2,\varnothing}^{k+1}(s)) = \cdots = \gamma^{n-1} V^{\pi^{k+1}}(s, \pi_{<n,\varnothing}^{k+1}(s)) \\
&= \gamma^{n-1}\left( r(s, \pi_{\leq n,\varnothing}^{k+1}(s)) + \gamma \sum_{s'} P(s'|s, \pi_{\leq n,\varnothing}^{k+1}(s)) V^{\pi^{k+1}}(s') \right) \\
&= \gamma^{n-1}\left( r(s, \pi_{\leq n,\varnothing}^{k+1}(s)) + \gamma \sum_{s'} P(s'|s, \pi_{\leq n,\varnothing}^{k+1}(s)) V^{\pi^{k}}(s') \right) \\
&= \gamma^{n-1} Q^{\pi^k}\left( s, \pi_{<n,\varnothing}^{k+1}(s); \pi_n^{k+1}(s, \pi_{<n,\varnothing}^{k+1}(s)) \right) \\
&\geq \gamma^{n-1} Q^{\pi^k}\left( s, \pi_{<n,\varnothing}^{k+1}(s); \pi_n^{k}(s, \pi_{<n,\varnothing}^{k+1}(s)) \right) \\
&= \gamma^{n-2} Q^{\pi^k}\left( s, \pi_{<n-1,\varnothing}^{k+1}(s); \pi_{n-1}^{k+1}(s, \pi_{<n-1,\varnothing}^{k+1}(s)) \right) \\
&\geq \cdots \geq \gamma^{i-1} Q^{\pi^k}(s, \pi_{<i,\varnothing}^{k+1}(s); \pi_i^{k+1}(s, \pi_{<i,\varnothing}^{k+1}(s))) \\
&> \gamma^{i-1} Q^{\pi^k}(s, \pi_{<i,\varnothing}^{k}(s); \pi_i^{k}(s, \pi_{<i,\varnothing}^{k}(s))) \\
&= \gamma^{i-1} V^{\pi^k}(s, \pi_{<i,\varnothing}^{k}(s)) = V^{\pi^k}(s).
\end{aligned}
\tag{50}
$$

The first equality follows from Proposition B.1 (ii). The third equality uses the fact that $V^{\pi^k}(s)$ has already converged. The strict inequality follows from the update rule, which preferentially selects the pre-update policy.

Hence $V^{\pi^{k+1}}(s) > V^{\pi^k}(s)$, contradicting the assumption that $V^{\pi^k}(s)$ has converged. Therefore, $\pi_{\mathcal{N},\varnothing}^k(s) = \pi_{\mathcal{N},\varnothing}^{k+1}(s)$ for $k \geq M$, implying that the joint policy $\pi_{\mathcal{N},\varnothing}^k(s)$ also converges. $\qquad\square$

In order to deduce the convergence of individual policies from the convergence of $\pi_{\mathcal{N},\varnothing}^k(s)$, we employ induction. To make the induction work properly, we need to construct a special ordering.

**Definition D.2.** Let $\mathcal{C} = \mathcal{P}(\{1, 2, \ldots, n-1\})$, where $\mathcal{P}$ denotes the power set. We introduce a binary relation $<$ on $\mathcal{C}$ as follows:

$$
A < B \Longleftrightarrow \min(A \setminus B) > \min(B \setminus A), \ A, B \in \mathcal{C}.
\tag{51}
$$

For the case involving the empty set, we define $\min \varnothing := n$.

**Lemma D.3.** *For any $A, B \in \mathcal{C}$, the following holds:*

$$
A < B \Longleftrightarrow \min(A \Delta B) \in B \setminus A,
$$

*where $\Delta$ denotes the symmetric difference, i.e., $A \Delta B := (A \setminus B) \cup (B \setminus A)$.*

*Proof.* We first prove the direction "$\Leftarrow$". Let $k = \min(A \Delta B) \in B \setminus A$. Since $A \setminus B \subseteq A \Delta B$, we have $\min(A \Delta B) \leq \min(A \setminus B)$. Moreover, because $k \in B \setminus A$, it follows that $\min(A \Delta B) = \min(B \setminus A)$. Since $(B \setminus A) \cap (A \setminus B) = \varnothing$, we obtain $\min(A \Delta B) < \min(A \setminus B)$. Thus, $\min(B \setminus A) < \min(A \setminus B)$, i.e., $A < B$.

Conversely, assume $A < B$. Then $\min(B \setminus A) < \min(A \setminus B)$. Hence,

$$
\min(A \Delta B) = \min\{\min(B \setminus A), \min(A \setminus B)\} = \min(B \setminus A).
\tag{52}
$$

Therefore, $\min(A \Delta B) \in B \setminus A$. $\qquad\square$

**Proposition D.4.** *The binary relation $<$ on $\mathcal{C}$ is a strict total order.*

*Proof.* Irreflexivity and asymmetry are immediate. We now prove that $<$ is connected and transitive.

*(Connectedness):* Let $A, B \in \mathcal{C}$ with $A \neq B$. We distinguish three cases:

(i) If $A \subsetneq B$, then $B \setminus A \neq \varnothing$. Hence $\min(B \setminus A) \leq n-1$ and $\min(A \setminus B) = n > \min(B \setminus A)$, so $A < B$.

(ii) If $B \subsetneq A$, by symmetry we obtain $B < A$.

(iii) If $A \nsubseteq B$ and $B \nsubseteq A$, then $\min(A \setminus B) \neq \min(B \setminus A)$. Thus, either $A < B$ or $B < A$.

*(Transitivity):* Let $A, B, C \in \mathcal{C}$ with $A < B$ and $B < C$.

If $A = \varnothing$, then clearly $A < C$. Otherwise, set $k_1 = \min(A \Delta B)$ and $k_2 = \min(B \Delta C)$. By Lemma D.3, we have $k_1 \in B \setminus A$ and $k_2 \in C \setminus B$. We analyze three cases:

(i) If $k_1 < k_2$, then $k_1 \notin B \Delta C$. Since $k_1 \in B$, it follows that $k_1 \in B \cap C$. Moreover, as $k_1 \in B \setminus A$, we have $k_1 \in C \setminus A$. Now, we only need to show $k_1 = \min(A \Delta C)$. For all $i < k_1$ with $i \in A \cup B$, we have $i \notin A \Delta B$, hence $i \in A \cap B$. Since $k_1 < k_2$, we also get $i \in A \cap B \cap C$, implying $i \notin A \Delta C$. Thus $k_1 = \min(A \Delta C)$, and by Lemma D.3, $A < C$.

(ii) If $k_2 < k_1$, then by symmetry, $k_2 = \min(A \Delta C)$ and $k_2 \in C \setminus A$. Hence $A < C$.

(iii) If $k_1 = k_2$, then $k_1 \in B \setminus A$ and $k_1 \in C \setminus B$ simultaneously, which is a contradiction. Thus this case cannot occur.

Therefore, $<$ is transitive. $\square$

After defining the strict total order $<$, we arrange the elements of $\mathcal{C}$ in ascending order as $\mathcal{C} = \{C_1, C_2, \ldots, C_{|\mathcal{C}|}\}$.

**Lemma D.5.** *Let $C_m \in \mathcal{C}$ and $C_m \neq \varnothing$. Let $G_d$ be the ADG of $\pi$ under state $s$. If $N_d(i) \nsupseteq C_m, i \in \mathcal{N}$, denote $k = \min(C_m \setminus N_d(i))$, and define*

$$C_j = A \cup B := \{x \in C_m \mid x < k\} \cup \{x \in \{1, \ldots, n-1\} \mid x > k\}. \tag{53}$$

*Then we have $j < m$. Furthermore, if $i \neq k$, the following holds:*

$$\pi_{i, C_j}(s, \pi_{C_j, C_m}(s, a_{C_m})) = \pi_{i, C_m}(s, a_{C_m}). \tag{54}$$

*Proof.* We first verify that $j < m$. Since $N_d(i) \nsupseteq C_m$, we have $C_m \setminus N_d(i) \neq \varnothing$, and hence $k < n$. By construction of $C_j$, $\min(C_m \setminus C_j) = k$. Therefore,

$$\min(C_j \Delta C_m) = \min\{\min(C_j \setminus C_m), \min(C_m \setminus C_j)\} = \min\{\min(C_j \setminus C_m), k\}. \tag{55}$$

Because $\min(C_j \setminus C_m) > k$, we obtain $\min(C_j \Delta C_m) = k \in C_m \setminus C_j$. By Lemma D.3, this implies $C_j < C_m$, i.e., $j < m$.

When $i \in C_m$, since $i \neq k$, we have $i \in C_m \cap C_j$, and therefore

$$\pi_{i, C_j}(s, \pi_{C_j, C_m}(s, a_{C_m})) = a_i = \pi_{i, C_m}(s, a_{C_m}). \tag{56}$$

When $i \notin C_m$, we consider the two cases $k = 1$ and $k > 1$ respectively.

(i) $k = 1$.

In this case, $C_j = \{2, 3, \ldots, n-1\}$. We analyze the construction of $\pi_{i, C_j}(s, a_{C_j})$:

$$\begin{aligned}
\pi_{i, C_j}(s, a_{C_j}) &= \pi_i(s, \pi_{(<i), C_j}(s, a_{C_j})) \\
&= \pi_i(s, \pi_{1, C_j}(s, a_{C_j}), \pi_{(<i) \cap C_j, C_j}(s, a_{C_j})) \\
&= \pi_i(s, \pi_1(s), a_{(<i) \cap C_j}) \\
&= \pi_i(s, \pi_1(s), a_{(<i) \cap (C_m \setminus \{1\})}, a_{(<i) \setminus C_m}).
\end{aligned} \tag{57}$$

Substituting $\pi_{C_j, C_m}(s, a_{C_m})$ into $a_{C_j}$, we obtain

$$\begin{aligned}
&\pi_{i, C_j}(s, \pi_{C_j, C_m}(s, a_{C_m})) \\
&= \pi_i(s, \pi_{(<i), C_j}(s, a_{C_j}))\big|_{a_{C_j} = \pi_{C_j, C_m}(s, a_{C_m})} \\
&= \pi_i(s, \pi_1(s), a_{(<i) \cap (C_m \setminus \{1\})}, \pi_{(<i) \setminus C_m, C_m}(s, a_{C_m})).
\end{aligned} \tag{58}$$

Since $1 \notin N_d(i)$, $\pi_i$ does not depend on $a_1$. Thus, replacing $\pi_1(s)$ with $a_1$, we obtain

$$\begin{aligned}
&\pi_{i, C_j}(s, \pi_{C_j, C_m}(s, a_{C_m})) \\
&= \pi_i(s, a_1, a_{(<i) \cap (C_m \setminus \{1\})}, \pi_{(<i) \setminus C_m, C_m}(s, a_{C_m})) \\
&= \pi_i(s, a_{(<i) \cap C_m}, \pi_{(<i) \setminus C_m, C_m}(s, a_{C_m})) \\
&= \pi_{i, C_m}(s, a_{C_m}).
\end{aligned} \tag{59}$$

(ii) $k > 1$.

We proceed by induction to prove that

$$\pi_{\ell,C_j}(s, \pi_{C_j,C_m}(s, a_{C_m})) = \pi_{\ell,C_m}(s, a_{C_m}), \quad \forall\, 1 \leq \ell < k. \tag{60}$$

For $\ell = 1$, since $1 < k$, we have $1 \in C_m \cap C_j$. Thus,

$$\pi_{1,C_j}(s, \pi_{C_j,C_m}(s, a_{C_m})) = a_1 = \pi_{1,C_m}(s, a_{C_m}). \tag{61}$$

Assume the statement (60) holds for all indices less than $\ell$. For $\ell < k$, note that $(< \ell) \cap C_m = (< \ell) \cap C_j$ and $(< \ell) \setminus C_m = (< \ell) \setminus C_j$. Therefore,

$$\pi_{\ell,C_j}(s, \pi_{C_j,C_m}(s, a_{C_m}))$$
$$= \pi_\ell(s, a_{(<\ell)\cap C_j}, \pi_{(<\ell)\setminus C_j, C_j}(s, a_{C_j}))\big|_{a_{C_j} = \pi_{C_j,C_m}(s, a_{C_m})} \tag{62}$$
$$= \pi_\ell(s, a_{(<\ell)\cap C_m}, \pi_{(<\ell)\setminus C_m, C_j}(s, \pi_{C_j,C_m}(s, a_{C_m}))).$$

By the induction hypothesis,

$$\pi_{(<\ell)\setminus C_m, C_j}(s, \pi_{C_j,C_m}(s, a_{C_m})) = \pi_{(<\ell)\setminus C_m, C_m}(s, a_{C_m}), \tag{63}$$

which yields

$$\pi_{\ell,C_j}(s, \pi_{C_j,C_m}(s, a_{C_m})) = \pi_\ell(s, a_{(<\ell)\cap C_m}, \pi_{(<\ell)\setminus C_m, C_m}(s, a_{C_m})) = \pi_{\ell,C_m}(s, a_{C_m}). \tag{64}$$

Finally, similar to (58), analyzing the construction of $\pi_{i,C_j}(s, a_{C_j})$ gives

$$\pi_{i,C_j}(s, \pi_{C_j,C_m}(s, a_{C_m}))$$
$$= \pi_i(s, \pi_{<i,C_j}(s, a_{C_j}))\big|_{a_{C_j} = \pi_{C_j,C_m}(s, a_{C_m})}$$
$$= \pi_i(s, a_{(<i)\cap C_m}, \pi_{(<k)\setminus C_m, C_j}(s, \pi_{C_j,C_m}(s, a_{C_m})), \pi_{k,C_j}(s, \pi_{C_j,C_m}(s, a_{C_m})), \pi_{C_j\setminus C_m, C_m}(s, a_{C_m})). \tag{65}$$

Since $k \notin N_d(i)$, we can replace $\pi_{k,C_j}(s, \pi_{C_j,C_m}(s, a_{C_m}))$ by $\pi_{k,C_m}(s, a_{C_m})$, yielding

$$\pi_{i,C_j}(s, \pi_{C_j,C_m}(s, a_{C_m}))$$
$$= \pi_i(s, a_{(<i)\cap C_m}, \pi_{(<k)\setminus C_m, C_m}(s, a_{C_m}), \pi_{k,C_m}(s, a_{C_m}), \pi_{C_j\setminus C_m, C_m}(s, a_{C_m}))$$
$$= \pi_i(s, a_{(<i)\cap C_m}, \pi_{(<i)\setminus C_m, C_m}(s, a_{C_m})) \tag{66}$$
$$= \pi_{i,C_m}(s, a_{C_m}).$$

$\square$

**Lemma D.6.** *Let $\{\pi^k\}_{k=1}^\infty$ be the policy sequence generated by Algorithm 1. Then for every $C_m \in \mathcal{C}$, $\pi_{\mathcal{N},C_m}^k$ converges within a finite number of steps.*

*Proof.* We proceed by induction.

**Base case:** Consider $C_1 = \varnothing$. By Lemma D.1, we directly obtain that $\pi_{\mathcal{N},C_1}^k$ converges in finitely many steps.

**Induction step:** Assume that $\pi_{\mathcal{N},C_j}^k$ has already converged for all $j < m$ when $k \geq M$. We now prove that $\pi_{\mathcal{N},C_m}^k$ also converges in finitely many steps.

We first show that, when $k \geq M$, for any $\max C_m < i \leq n$, the following inequality holds:

$$Q^{\pi^k}(s, \pi_{<i,C_m}^{k+1}(s, a_{C_m}); \pi_{i,C_m}^{k+1}(s, a_{C_m})) \geq \gamma^{-1} Q^{\pi^k}(s, \pi_{<i-1,C_m}^{k+1}(s, a_{C_m}); \pi_{i-1,C_m}^{k+1}(s, a_{C_m})). \tag{67}$$

(i): If $N_d(i) \supseteq C_m$, then by the update rule,

$$Q^{\pi^k}(s, \pi_{<i,C_m}^{k+1}(s, a_{C_m}); \pi_{i,C_m}^{k+1}(s, a_{C_m}))$$
$$= Q^{\pi^k}(s, \pi_{<i,C_m}^{k+1}(s, a_{C_m}); \pi_i^{k+1}(s, \pi_{<i,C_m}^{k+1}(s, a_{C_m})))$$
$$\geq Q^{\pi^k}(s, \pi_{<i,C_m}^{k+1}(s, a_{C_m}); \pi_i^k(s, \pi_{<i,C_m}^{k+1}(s, a_{C_m}))) \tag{68}$$
$$= \gamma^{-1} Q^{\pi^k}(s, \pi_{<i-1,C_m}^{k+1}(s, a_{C_m}); \pi_{i-1}^{k+1}(s, \pi_{<i-1,C_m}^{k+1}(s, a_{C_m}))).$$

(ii): If $N_d(i) \not\supseteq C_m$, we then construct $C_j = \{x \in C_m \mid x < k'\} \cup \{x \in \{1, \ldots, n-1\} \mid x > k'\}$ with $k' = \min(C_m \setminus N_d(i))$ according to Lemma D.5. Since $i > \max C_m$, we have $i \neq k'$, and therefore $\pi_{i,C_m}^k(s, a_{C_m}) = \pi_{i,C_j}^k(s, \pi_{C_j,C_m}^k(s, a_{C_m}))$. By the induction hypothesis, $\pi_{\mathcal{N},C_j}^k$ has already converged. Hence,

$$
\begin{aligned}
\pi_{i,C_m}^{k+1}(s, a_{C_m}) &= \pi_{i,C_j}^{k+1}(s, \pi_{C_j,C_m}^{k+1}(s, a_{C_m})) \\
&= \pi_{i,C_j}^k(s, \pi_{C_j,C_m}^{k+1}(s, a_{C_m})) \\
&= \pi_i^k(s, \pi_{<i,C_j}^k(s, a_{C_j}))\big|_{a_{C_j} = \pi_{C_j,C_m}^{k+1}(s, a_{C_m})} \\
&= \pi_i^k(s, \pi_{<i,C_j}^k(s, \pi_{C_j,C_m}^{k+1}(s, a_{C_m}))).
\end{aligned}
\tag{69}
$$

We examine each component of $\pi_{<i,C_j}^{k+1}(s, \pi_{C_j,C_m}^{k+1}(s, a_{C_m}))$. If $x \in (< i)$ and $x = k'$, since $k' \notin N_d(i)$, then $\pi_i^k$ is independent of $a_x$, so we replace $\pi_{x,C_j}^k(s, \pi_{C_j,C_m}^{k+1}(s, a_{C_m}))$ with $\pi_{x,C_m}^{k+1}(s, a_{C_m})$ in (69). If $x \neq k'$, then we again apply Lemma D.5 and the induction hypothesis to obtain

$$
\pi_{x,C_j}^k(s, \pi_{C_j,C_m}^{k+1}(s, a_{C_m})) = \pi_{x,C_j}^{k+1}(s, \pi_{C_j,C_m}^{k+1}(s, a_{C_m})) = \pi_{x,C_m}^{k+1}(s, a_{C_m}).
\tag{70}
$$

Thus,

$$
\pi_{i,C_m}^{k+1}(s, a_{C_m}) = \pi_i^k(s, \pi_{<i,C_j}^k(s, \pi_{C_j,C_m}^{k+1}(s, a_{C_m}))) = \pi_i^k(s, \pi_{<i,C_m}^{k+1}(s, a_{C_m})).
\tag{71}
$$

Therefore,

$$
\begin{aligned}
&Q^{\pi^k}(s, \pi_{<i,C_m}^{k+1}(s, a_{C_m}); \pi_{i,C_m}^{k+1}(s, a_{C_m})) \\
&= Q^{\pi^k}(s, \pi_{<i,C_m}^{k+1}(s, a_{C_m}); \pi_i^k(s, \pi_{<i,C_m}^{k+1}(s, a_{C_m}))) \\
&= \gamma^{-1} Q^{\pi^k}(s, \pi_{<i-1,C_m}^{k+1}(s, a_{C_m}); \pi_{i-1}^{k+1}(s, \pi_{<i-1,C_m}^{k+1}(s, a_{C_m}))).
\end{aligned}
\tag{72}
$$

Next, we prove

$$
Q^{\pi^k}(s, \pi_{<\max C_m, C_m}^{k+1}(s, a_{C_m}); \pi_{\max C_m, C_m}^{k+1}(s, a_{C_m})) = Q^{\pi^k}(s, \pi_{<\max C_m, C_m}^k(s, a_{C_m}); \pi_{\max C_m, C_m}^k(s, a_{C_m})).
\tag{73}
$$

We examine each component of $\pi_{\leq \max C_m, C_m}^{k+1}(s, a_{C_m})$.

(iii): If $i \in C_m$, then $\pi_{i,C_m}^k(s, a_{C_m}) = a_i = \pi_{i,C_m}^{k+1}(s, a_{C_m})$.

(iv): If $i \notin C_m$, let $C_j = C_m \cap (< i)$. Since $i < \max C_m$, we have $C_j \subsetneq C_m$, and by Definition D.2, $j < m$. By the induction hypothesis, $\pi_{i,C_j}^k$ has converged. As $\pi_{\leq i}^k$ depends only on the first $i-1$ agents, it follows that $\pi_{i,C_m}^k = \pi_{i,C_j}^k$. Therefore,

$$
\pi_{i,C_m}^k(s, a_{C_m}) = \pi_{i,C_j}^k(s, a_{C_j}) = \pi_{i,C_j}^{k+1}(s, a_{C_j}) = \pi_{i,C_m}^{k+1}(s, a_{C_m}).
\tag{74}
$$

Thus, $\pi_{\leq \max C_m, C_m}^{k+1}(s, a_{C_m}) = \pi_{\leq \max C_m, C_m}^k(s, a_{C_m})$, and hence (73) holds.

Now,

$$
\begin{aligned}
&Q^{\pi^{k+1}}(s, \pi_{<n,C_m}^{k+1}(s, a_{C_m}); \pi_{n,C_m}^{k+1}(s, a_{C_m})) \\
&= Q^{\pi^k}(s, \pi_{<n,C_m}^{k+1}(s, a_{C_m}); \pi_{n,C_m}^{k+1}(s, a_{C_m})) \\
&\geq \cdots \geq \gamma^{\max C_m - n} Q^{\pi^k}(s, \pi_{<\max C_m, C_m}^{k+1}(s, a_{C_m}); \pi_{\max C_m, C_m}^{k+1}(s, a_{C_m})) \\
&= \gamma^{\max C_m - n} Q^{\pi^k}(s, \pi_{<\max C_m, C_m}^k(s, a_{C_m}); \pi_{\max C_m, C_m}^k(s, a_{C_m})) \\
&= Q^{\pi^k}(s, \pi_{<n,C_m}^k(s, a_{C_m}); \pi_{n,C_m}^k(s, a_{C_m})).
\end{aligned}
\tag{75}
$$

Here, the second line follows from Lemma D.1, which ensures that $V^{\pi^k}(s)$ has converged, and hence $Q^{\pi^k}(s, a)$ have converged; the third line follows from (67); the fourth line from (73).

Equation (75) shows that $Q^{\pi^k}(s, \pi_{<n,C_m}^k(s, a_{C_m}); \pi_{n,C_m}^k(s, a_{C_m}))$ is monotonically non-decreasing, and thus converges in finitely many steps.

Let $k \geq M' \geq M$ be such that $Q^{\pi^k}(s, \pi^k_{<n,C_m}(s, a_{C_m}); \pi^k_{n,C_m}(s, a_{C_m}))$ has converged. We now prove by contradiction that $\pi^k_{\mathcal{N},C_m}$ must also converge.

Suppose for $k \geq M'$, $\pi^k_{\mathcal{N},C_m}(s) \neq \pi^{k+1}_{\mathcal{N},C_m}(s)$. Let $i$ be the smallest index where they differ, i.e.,

$$\pi^k_{i,C_m}(s) \neq \pi^{k+1}_{i,C_m}(s), \quad \pi^k_{<i,C_m}(s) = \pi^{k+1}_{<i,C_m}(s). \tag{76}$$

By the analyses in (iii) and (iv), $i$ must satisfy $i > \max C_m$. If $N_d(i) \not\supseteq C_m$, then by (ii),

$$\pi^{k+1}_{i,C_m}(s, a_{C_m}) = \pi^k_i(s, \pi^{k+1}_{<i,C_m}(s, a_{C_m})) = \pi^k_i(s, \pi^k_{<i,C_m}(s, a_{C_m})) = \pi^k_{i,C_m}(s, a_{C_m}), \tag{77}$$

contradicting $\pi^k_{i,C_m}(s) \neq \pi^{k+1}_{i,C_m}(s)$. Therefore, it must be that $N_d(i) \supseteq C_m$.

$$
\begin{aligned}
&Q^{\pi^{k+1}}(s, \pi^{k+1}_{<n,C_m}(s, a_{C_m}); \pi^{k+1}_{n,C_m}(s, a_{C_m})) \\
&= Q^{\pi^k}(s, \pi^{k+1}_{<n,C_m}(s, a_{C_m}); \pi^{k+1}_{n,C_m}(s, a_{C_m})) \\
&\geq \gamma^{i-n} Q^{\pi^k}(s, \pi^{k+1}_{<i,C_m}(s, a_{C_m}); \pi^{k+1}_{i,C_m}(s, a_{C_m})) \\
&= \gamma^{i-n} Q^{\pi^k}(s, \pi^k_{<i,C_m}(s, a_{C_m}); \pi^{k+1}_i(s, \pi^k_{<i,C_m}(s, a_{C_m}))) \\
&> \gamma^{i-n} Q^{\pi^k}(s, \pi^k_{<i,C_m}(s, a_{C_m}); \pi^k_i(s, \pi^k_{<i,C_m}(s, a_{C_m}))) \\
&= Q^{\pi^k}(s, \pi^k_{<n,C_m}(s, a_{C_m}); \pi^k_{n,C_m}(s, a_{C_m})).
\end{aligned}
\tag{78}
$$

The second line uses the fact that $Q^{\pi^k}(s, a)$ has converged; the third line is by the update rule; the fifth line follows from the update rule, which preferentially selects the pre-update policy.

Equation (78) contradicts the fact that $Q^{\pi^k}(s, \pi^k_{<n,C_m}(s, a_{C_m}); \pi^k_{n,C_m}(s, a_{C_m}))$ has already converged. Therefore, for all $k \geq M'$, $\pi^k_{\mathcal{N},C_m}(s) = \pi^{k+1}_{\mathcal{N},C_m}(s)$, i.e., $\pi^k_{\mathcal{N},C_m}$ converges in finitely many steps. $\qquad\square$

**Proof of Theorem 5.1**

*Proof.* According to Lemma D.6, we have that $\pi^k_i = \pi^k_{i,<i}$ converges in a finite number of steps. Let $\pi$ be the limit point of the sequence $\{\pi^k\}_{k=1}^{\infty}$. From the update rule, it follows that

$$Q^{\pi}(s, \pi_{(<i),N_d(i)}(s, a_{N_d(i)}); \pi_{i,N_d(i)}(s, a_{N_d(i)})) = \max_{a_i} Q^{\pi}(s, \pi_{(<i),N_d(i)}(s, a_{N_d(i)}); a_i). \tag{79}$$

Therefore, the limit point of $\{\pi^k\}_{k=1}^{\infty}$ is a $G_d$-locally optimal policy. $\qquad\square$

**Proof of Corollary 5.3**

*Proof.* From the analysis in Lemma D.1, we know that $V^{\pi^k}(s)$ is monotonically non-decreasing. Hence, $V^{\pi^k}(s)$ also converges within a finite number of steps.

Suppose that for $k \geq M$, $V^{\pi^k}(s)$ has already converged. Assume further that $G_c(s, \hat{\pi}^{k+1}) \neq G_c(s, \hat{\pi}^k)$ when $k \geq M$. This implies $\hat{\pi}^{k+1} \neq \hat{\pi}^k$. From the contradiction argument in Lemma D.1, it follows that

$$V^{\pi^{k+1}}(s) = V^{\hat{\pi}^{k+1}}(s) > V^{\hat{\pi}^k}(s) = V^{\pi^k}(s), \tag{80}$$

which contradicts the assumption that $V^{\pi^k}(s)$ has already converged. Therefore, it must hold that $G_c(s, \hat{\pi}^{k+1}) = G_c(s, \hat{\pi}^k)$ for all $k \geq M$.

Consequently, when $k \geq M$, the dynamic CG eventually stabilizes into a static CG. During this stabilization stage, the update process reduces to Algorithm 1. By Theorem 5.1, $\pi^k$ converges within a finite number of steps. Let $\pi$ be the limit point of $\{\pi^k\}_{k=1}^{\infty}$. Then $\pi$ is a $G_d$-locally optimal policy. Furthermore, since the ADG satisfies condition (11), it follows from Theorem 4.4 that $\pi$ is globally optimal. $\qquad\square$

## E  CONSTRUCTION OF ACTION DEPENDENCY GRAPHS

To elucidate the construction of ADGs, we present Algorithm 2 that efficiently derives an ADG from a given CG such that the condition (11) is satisfied.

---

**Algorithm 2** Greedy Algorithm: Finding a Sparse ADG

---

  **Input:** A CG $G_c$
  **Output:** An ADG $G_d = (\mathcal{N}, \mathcal{E}_d)$
  Initialize an empty graph $G_d = (\mathcal{N}, \mathcal{E}_d)$ with all vertices unindexed
  **for** $i = 0$ **to** $n - 1$ **do**
      Assign new index $n-i$ to a vertex among the unindexed ones, such that the size of $N_c(\geq (n-i))$ is minimized
  **end for**
  Construct the edge set $\mathcal{E}_d$ by adding edges $(j, i)$ for each vertex $i \in \mathcal{N}$ and each $j \in N_c(\geq i)$ as specified in (11)

---

## F  EXPERIMENTAL DETAILS

**Setup of Coordination Polymatrix Game.** In matrix cooperative games, different CGs and their corresponding sparse ADGs are shown in Figure 9. For policy iteration methods, we randomly generate the parameters of the payoff matrices, while fixing the maximum reward to be equal to the number of edges in the CG. Moreover, the maximum reward is obtained when all agents choose action 1. For the MAPPO method, we use fixed payoff matrices, with the exact parameters provided in Table 2 to Table 5.

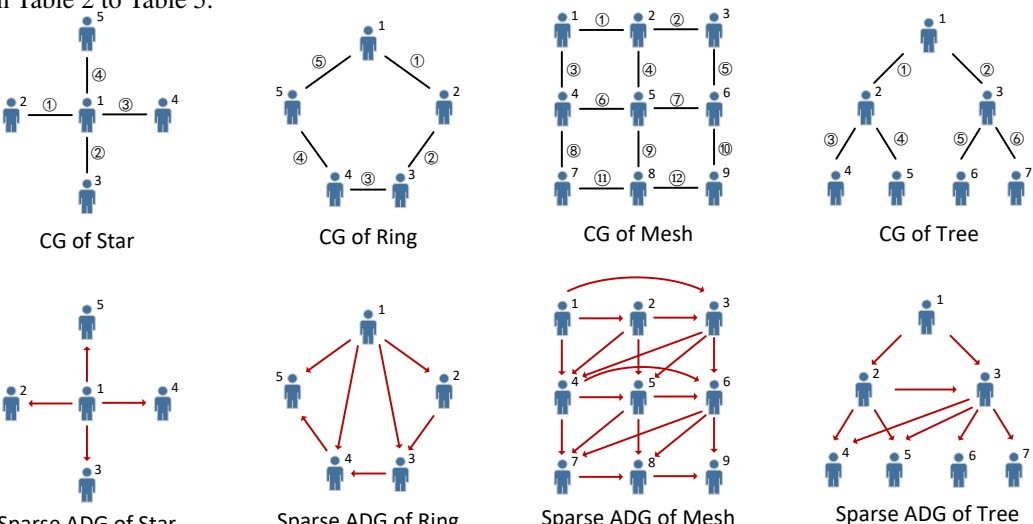

Figure 9: The CG and sparse ADG of polymatrix coordination game.

Table 2: ① in Star

Table 3: ② in Star

| $a_1 \backslash a_2$ | 0 | 1 | 2 | 3 | 4 |
|---|---|---|---|---|---|
| 0 | 3.5 | 5.0 | 0.5 | 0.5 | 0.5 |
| 1 | 0.5 | 3.5 | 6.0 | 0.5 | 0.5 |
| 2 | 0.5 | 0.5 | 3.5 | 0.5 | 0.5 |
| 3 | 0.5 | 0.5 | 0.5 | 3.25 | 0.5 |
| 4 | 0.5 | 0.5 | 0.5 | 0.5 | 3.0 |

| $a_1 \backslash a_3$ | 0 | 1 | 2 | 3 | 4 |
|---|---|---|---|---|---|
| 0 | 3.5 | 0.5 | 5.0 | 0.5 | 0.5 |
| 1 | 0.5 | 3.5 | 6.0 | 0.5 | 0.5 |
| 2 | 0.5 | 0.5 | 3.5 | 0.5 | 0.5 |
| 3 | 0.5 | 0.5 | 0.5 | 3.25 | 0.5 |
| 4 | 0.5 | 0.5 | 0.5 | 0.5 | 3.0 |

**Setup of ATSC.** In the ATSC environment, we conduct experiments on the maps `2x2grid`, `3x3grid`, and `RESCO/grid4x4` provided by SUMO-RL. Based on the road network connectivity,

Table 4: ③ in Star

| $a_1 \backslash a_4$ | 0 | 1 | 2 | 3 | 4 |
|---|---|---|---|---|---|
| 0 | 3.5 | 0.5 | 0.5 | 5.0 | 0.5 |
| 1 | 0.5 | 3.5 | 6.0 | 0.5 | 0.5 |
| 2 | 0.5 | 0.5 | 3.5 | 0.5 | 0.5 |
| 3 | 0.5 | 0.5 | 0.5 | 3.25 | 0.5 |
| 4 | 0.5 | 0.5 | 0.5 | 0.5 | 3.0 |

Table 5: ④ in Star

| $a_1 \backslash a_5$ | 0 | 1 | 2 | 3 | 4 |
|---|---|---|---|---|---|
| 0 | 3.5 | 0.5 | 0.5 | 0.5 | 5.0 |
| 1 | 0.0 | 0.0 | 0.0 | 0.0 | 0.0 |
| 2 | 0.5 | 0.5 | 3.5 | 0.5 | 0.5 |
| 3 | 0.5 | 0.5 | 0.5 | 3.25 | 0.5 |
| 4 | 0.5 | 0.5 | 0.5 | 0.5 | 3.0 |

we design corresponding CGs, with their adjacency lists for CGs and sparse ADGs reported in Table 6 to Table 11. To increase task difficulty and highlight the benefits of cooperation, we follow the idea of Li et al. (2021); Böhmer et al. (2020) to modify the reward function. Specifically, instead of assigning each agent its own queue reward, we redefine the reward as the minimum individual reward among its CG neighbors, thereby emphasizing the performance gap induced by cooperation.

Table 6: CG of 2x2grid

| Vertex | Neighbors |
|---|---|
| 0 | 1, 2 |
| 1 | 0, 3 |
| 2 | 0, 3 |
| 3 | 1, 2 |

Table 7: Sparse ADG of 2x2grid

| Vertex | Parent nodes |
|---|---|
| 0 | |
| 1 | 0 |
| 2 | 1, 0 |
| 3 | 2, 1 |

Table 8: CG of 3x3grid

| Vertex | Neighbors |
|---|---|
| 0 | 1, 3 |
| 1 | 0, 2, 4 |
| 2 | 1, 5 |
| 3 | 0, 4, 6 |
| 4 | 1, 3, 5, 7 |
| 5 | 2, 4, 8 |
| 6 | 3, 7 |
| 7 | 4, 6, 8 |
| 8 | 5, 7 |

Table 9: Sparse ADG of 3x3grid

| Vertex | Parent nodes |
|---|---|
| 0 | |
| 1 | 0 |
| 2 | 1, 0 |
| 3 | 2, 1, 0 |
| 4 | 3, 2, 1 |
| 5 | 4, 3, 2 |
| 6 | 5, 4, 3 |
| 7 | 6, 5, 4 |
| 8 | 7, 5 |

**Experimental Hyperparameters.** Our implementation of MAPPO is based on the open-source EPy-MARL framework Papoudakis et al. (2021), employing the Adam optimizer for training. We use the same hyperparameters across experiments under different ADGs, with a few critical hyperparameters adjusted to fit each environment. These modified values are reported in Table 14, while any unlisted parameters follow the default EPyMARL configuration.

**Neural Network Architecture.** For MAPPO with empty ADGs, we adopt the default MLP configurations specified in Papoudakis et al. (2021); Böhmer et al. (2020). For MAPPO with sparse and dense ADGs, we modify the agent network architecture as follows.

Let $o_i \in \mathbb{R}^{d_o}$ denote the observational features of agent $i$, and $a_i \in \mathbb{R}^{d_a}$ the action features of agent $i$. The first-layer hidden features are computed as:

$$h_{o_i}^1 = \text{ReLU}(W_1 o_i + b_1), \quad h_{a_i}^1 = \text{ReLU}(W_2 a_i + b_2),$$

where $W_1 \in \mathbb{R}^{64 \times d_o}$, $W_2 \in \mathbb{R}^{64 \times d_a}$ and $b_1, b_2 \in \mathbb{R}^{64}$ are the weights and biases, respectively.

Next, we take the average of $h_{a_i}^1$ over the dependency set $N_d(i)$:

$$h_{a_i}^2 = \frac{1}{|N_d(i)|} \sum_{i \in N_d(i)} h_{a_i}^1.$$

Table 10: CG of 4x4grid

| Vertex | Neighbors |
|--------|-----------|
| 0 | 1, 4 |
| 1 | 0, 2, 5 |
| 2 | 1, 3, 6 |
| 3 | 2, 7 |
| 4 | 0, 5, 8 |
| 5 | 1, 4, 6, 9 |
| 6 | 2, 5, 7, 10 |
| 7 | 3, 6, 11 |
| 8 | 4, 9, 12 |
| 9 | 5, 8, 10, 13 |
| 10 | 6, 9, 11, 14 |
| 11 | 7, 10, 15 |
| 12 | 8, 13 |
| 13 | 9, 12, 14 |
| 14 | 10, 13, 15 |
| 15 | 11, 14 |

Table 11: Sparse ADG of 4x4grid

| Vertex | Parent nodes |
|--------|--------------|
| 0 | |
| 1 | 0 |
| 2 | 0, 1 |
| 3 | 0, 1, 2 |
| 4 | 0, 1, 2, 3 |
| 5 | 1, 2, 3, 4 |
| 6 | 2, 3, 4, 5 |
| 7 | 3, 4, 5, 6 |
| 8 | 4, 5, 6, 7 |
| 9 | 5, 6, 7, 8 |
| 10 | 6, 7, 8, 9 |
| 11 | 7, 8, 9, 10 |
| 12 | 8, 9, 10, 11 |
| 13 | 9, 10, 11, 12 |
| 14 | 10, 11, 13 |
| 15 | 11, 14 |

Table 12: CG of Aloha

| Vertex | Neighbors |
|--------|-----------|
| 0 | 1, 5 |
| 1 | 0, 2, 6 |
| 2 | 1, 3, 7 |
| 3 | 2, 4, 8 |
| 4 | 3, 5, 9 |
| 5 | 0, 4, 6 |
| 6 | 1, 5, 7 |
| 7 | 2, 6, 8 |
| 8 | 3, 7, 9 |
| 9 | 4, 8 |

Table 13: Sparse ADG of Aloha

| Vertex | Parent nodes |
|--------|--------------|
| 0 | 1, 5 |
| 1 | 2, 6 |
| 2 | 3, 7 |
| 3 | 8, 4 |
| 4 | 9, 5 |
| 5 | 6 |
| 6 | 7 |
| 7 | 8 |
| 8 | 9 |
| 9 | |

Table 14: Experimental Hyperparameters

| | learning rate | weight decay | buffer size | batch size | entropy coefficient |
|--|---------------|--------------|-------------|------------|---------------------|
| ATSC | 0.0004 | 0.0001 | 8 | 8 | 0.02 |
| Polymatrix Game | 0.0004 | 0.0001 | 16 | 8 | 0.1 |
| Aloha | 0.0005 | 0.0001 | 16 | 16 | 0.01 |

We then concatenate the two feature vectors and obtain

$$h^3 = [h^1_{o_i}, h^2_{a_i}],$$

which is fed into a multilayer perceptron:

$$h^4 = \text{ReLU}(W_3 h^3 + b_3), \quad z = W_4 h^4 + b_4,$$

where $W_3 \in \mathbb{R}^{64 \times 128}$, $W_4 \in \mathbb{R}^{d_{\text{action}} \times 64}$, $b_3 \in \mathbb{R}^{64}$, and $b_4 \in \mathbb{R}^{d_{\text{action}}}$. The final output is $z$.

## G  ADDITIONAL EXPERIMENTS

### G.1  BASELINE COMPARISONS IN ATSC

We include additional comparisons against standard MARL baselines, including QMIX, COMA, and MAT, conducted in the ATSC environment. MAT (Wen et al., 2022) is an algorithm that employs

auto-regressive policies and models action dependencies through attention networks. By modifying the attention mask to suppress interactions outside the ADG, we construct a sparse-ADG variant of MAT (denoted MAT-sparse). The results are provided in Figure 10.

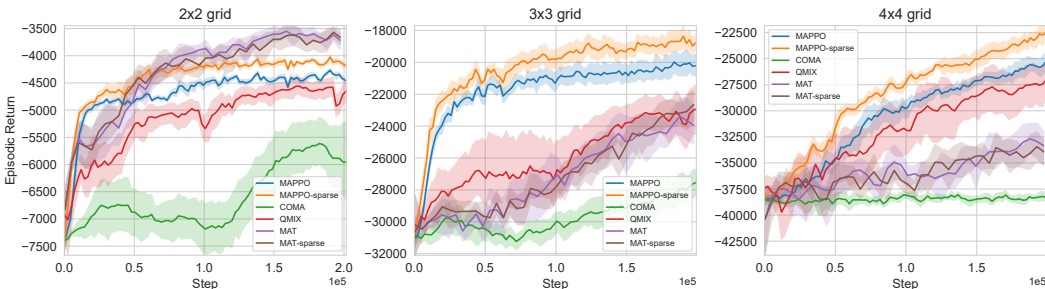

Figure 10: Results of ATSC with baselines.

The results show that the sparse-ADG version of MAPPO consistently outperforms its independent-policy counterpart. On 3x3 and 4x4 grids, MAT exhibits only moderate performance, likely due to its slower learning efficiency. Nevertheless, the learning curves of MAT-sparse and the original MAT remain closely aligned, suggesting that enforcing sparse ADGs does not introduce performance degradation for action-dependent policies.

## G.2 RESULTS OF THE ALOHA ENVIRONMENT

We further evaluate the algorithms in Aloha Oliehoek (2010). Aloha consists of 10 islands, each equipped with a radio tower that transmits messages to local residents. Each island maintains a message queue, and at every timestep an agent may choose to transmit a message or stay idle. Due to geographical proximity, simultaneous transmissions from adjacent islands interfere: when two neighboring agents transmit at the same time, a collision occurs and the messages must be resent. A successful transmission yields a global reward of 1 , while a collision incurs a penalty of -10.

We use the adjacency matrix provided by the environment as the CG and generate the corresponding sparse ADG (Table 13) using Algorithm 2 . Results are given in Figure 11 (left). Algorithms based on independent policies struggle to obtain positive transmission rewards, whereas ADG-based policies successfully learn efficient transmission policies.

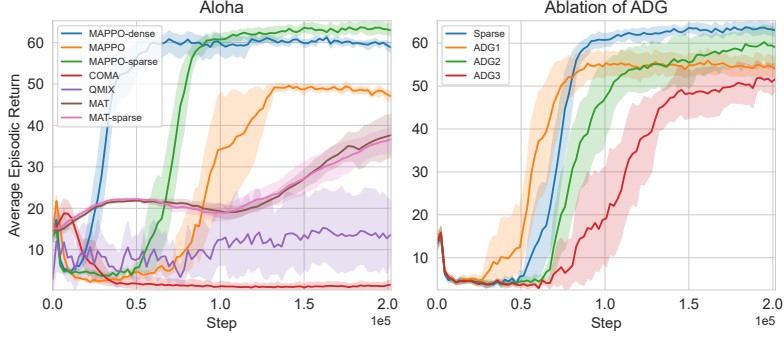

Figure 11: Results of Aloha (left) and Ablation on heuristic ADGs (right).

## G.3 ABLATION ON DIFFERENT ADGS

To examine the effect of alternative ADG constructions, particularly those that do not satisfy the structural conditions of Theorem 4.4, we conducted ablations in the Aloha environment. We designed several heuristic ADGs in which each agent depends only on the actions of the $k$ immediately preceding agents ($k = 1, 2, 3$). Notably, in the ADG generated by Algorithm 2, each node depends on at most 3 agents. The results, shown in Figure 11 (right), indicate that such heuristically constructed

ADGs perform noticeably worse than ADGs derived from the CG. This supports the importance of using CG-consistent dependency structures when CG is available.

Different agent index orderings lead to different ADG structures according to (11). To examine robustness with respect to ADG construction, we evaluate the sparse-ADG variant of MAPPO on 2x2 and 4x4 ATSC grids using both ascending and descending index orders. As shown in Figure 12, the training curves under different indexings are nearly identical, indicating that the algorithm is insensitive to index order.

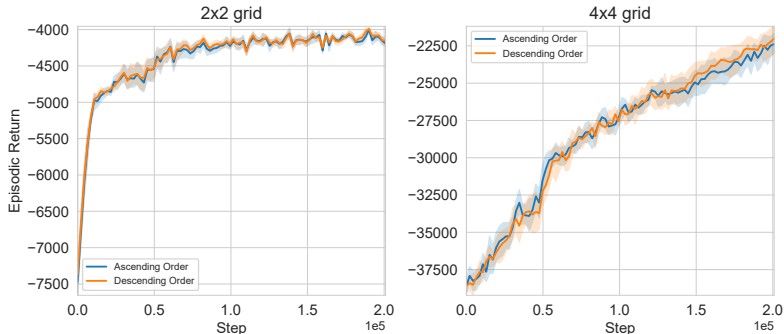

Figure 12: Results of Different Agent Index Orderings .

## G.4 RESULTS OF SMAC

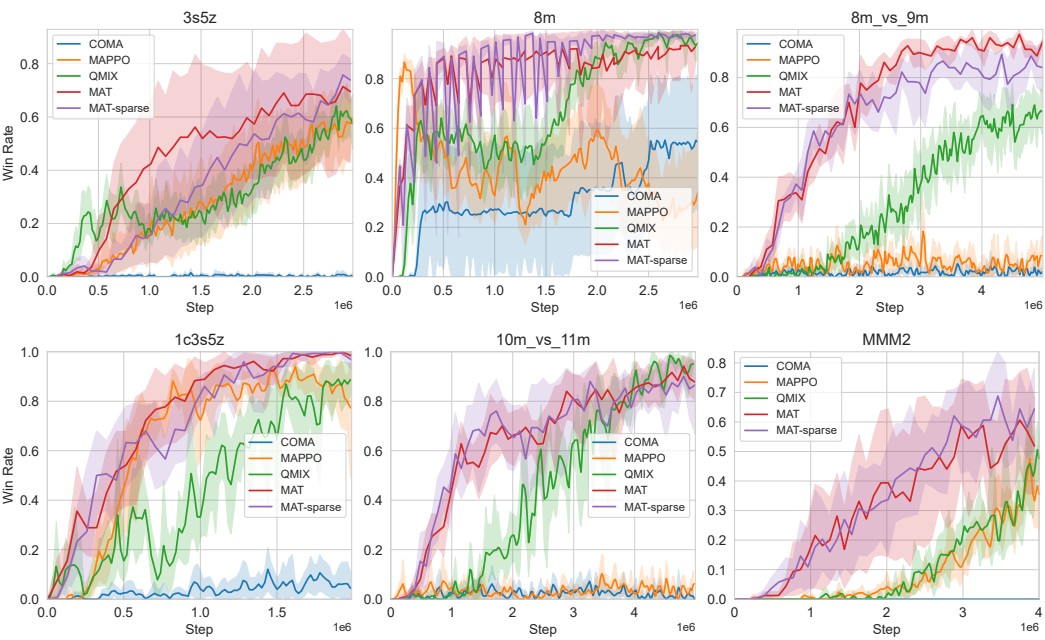

Figure 13: Results of SMAC.

To further assess the behavior of sparse ADGs in complex dynamic settings, we perform experiments in the SMAC environment. Since SMAC does not provide an explicit CG, we cannot construct ADGs guaranteed to satisfy the conditions in Theorem 4.4. Thus, sparse ADGs in this setting are not expected to match the performance of dense ADGs. We adopt a simple rule for sparse ADGs: each agent depend on at most 1/4 of the total number of agents. (e.g., in the 8m map, each agent only depends on the actions of its two immediate predecessors) These experiments are intended solely to evaluate whether sparse ADGs still outperform independent-policy baselines. We train MAT and its sparse-ADG variant on several SMAC maps. We adopt the hyperparameters

recommended by Wen et al. (2022) for the MAT and its sparse-ADG variant, while adopting the hyperparameters recommended by EPYMARL for the baseline algorithms. The results in Figure 13 show that sparse ADG policies still outperform independent ones, and does not exhibit significant degradation compared auto-regressive policies.

