# OpenReview forum: "Action Dependency Graphs for Globally Optimal Coordinated Reinforcement Learning"
_ICLR.cc/2026/Conference — Submitted to ICLR 2026_

### Official Review · Reviewer_4W8F · 2025-10-29

**Soundness:** 3
**Presentation:** 3
**Contribution:** 3
**Rating:** 6
**Confidence:** 4

**Summary:**

The paper considers the problem of optimizing action-dependent policies in MAMDPs where the global Q-value can be decomposed by a coordination graph (which can be policy- and state-dependent). The paper contributes a sufficient condition (regarding the action dependency graph) for the action-dependent policy to ensure the global optimality, along with a tabular algorithm to optimize such action-dependent policies.

**Strengths:**

The paper is exceptionally well-written with clear use of notions and illustrative diagrams (Figures 1-4) that help understanding.


The paper reveals an important link between coordination graphs and action-dependent policies, which has been a missing piece in the cooperative MARL literature.

**Weaknesses:**

1. While the clarity is overall good, there is room for improvement:
    - *Any* deterministic policy can be decomposed into independent policies as  $\pi(s)=(\pi_1(s),\ldots,\pi_n(s))$ can therefore some statements in the first two paragraphs of Section 4.1 are somewhat sloppy.
    - The notation used in (8) and (9) is hard to digest, which could benefit with more explanations and examples. Without understanding (8), it’s hard to understand (10).

2. A reader might appreciate a discussion on (the difficulty of) extending the idea and techniques to stochastic policies.


3. In Figure 5, it’s unclear how sparse the graphs generated by Algorithm 2 are. It’s unclear in general what kind (e.g., how spare) of graphs are generated by Algorithm 2 and how to let them satisfy (11).

**Questions:**

All my concerns are in the Weaknesses section.

---

> ### Author Response · Authors · 2025-11-26
> **Response 1 Reviewer 4W8F**
>
> We thank the reviewer for the constructive feedback and for highlighting several points that required clarification or further discussion. Below we provide detailed responses to each concern and describe the revisions made accordingly.
>
> **Weakness 1. Clarity improvement.**
>
> We revised the discussion in Section 4.1 to clarify the relationship between joint and independent policies:
> > In MARL, a deterministic joint policy $\pi(s)$ is a vector-valued mapping from states to actions, where each component corresponds to an individual agent's action. Thus, $\pi(s)$ can always be written as a collection of independent policies $(\pi_1(s), \pi_2(s), \dots, \pi_n(s))$. In this sense, the independent policies is expressive enough to represent any joint deterministic policy, including an optimal one. However, this does not imply that independent learning can converge to the optimal joint policy, since independent policies cannot capture coordinated behaviors that rely on correlated actions across agents. The absence of such correlations may cause independent learners to converge to suboptimal policies.
>
> We also added explanations between Eqs. (8) and (9).
> > The key difference between $\pi_{i,C}$ and action-dependent policy $\pi_{i}$ is that the former's actions in $(<i)\setminus C$ are already determined by $\pi_{<i,C}$, and therefore $\pi_{i,C}$ depends only on the actions in $C$. A special case is $\pi_{i,\varnothing}$, where no other agent's action is involved. In this case, $\pi_{i,\varnothing}$ reduces exactly to the standard independent policy.
>
>
> **Weakness 2. Discussion on (the difficulty of) extending the idea and techniques to stochastic policies.**
>
> We agree that extending ADGs to stochastic policies is both possible and nontrivial. Several open challenges remain:
>
> 1. The definition of a $G_d$-locally optimal point must be generalized to stochastic policies. Transitioning from deterministic to stochastic policies is analogous to moving from pure to mixed strategies in game theory, a shift that typically increases the number of equilibria. Similarly, adopting stochastic policies will introduce additional $G_d$-locally optimal points. It is therefore necessary to analyze whether these new optima continue to satisfy the optimality guarantees of Theorem 4.4.
>
> 2. Our current policy iteration procedure yields only deterministic policies, and thus cannot be used to study convergence properties of stochastic policies. Analyzing stochastic policies likely requires alternative optimization methods, such as policy gradient methods.
>
> 3. A deterministic action-dependent policy induces a deterministic joint policy, which can be decomposed into independent policies. This allows us to deploy the learned policies in a fully decentralized manner after training. In contrast, stochastic action-dependent policies may involve correlated action distributions that cannot be factorized into independent policies. As a result, stochastic ADG policies always require communication during execution, restricting their applicability to environments that support message passing.
>
>
> **Weakness 3. In Figure 5, it's unclear how sparse the graphs generated by Algorithm 2 are. It's unclear in general what kind (e.g., how spare) of graphs are generated by Algorithm 2 and how to let them satisfy (11).**
>
> 1. **What kind of graphs are generated by Algorithm 2**
>
> Algorithm 2 constructs an ADG whose number of directed edges is generally not smaller than the number of undirected edges in the underlying CG. This arises because an agent’s parents in the ADG are drawn from a sequence of its CG neighbors, so the sparsity of the ADG is largely inherited from the sparsity of the CG itself. When the CG is dense (e.g., a complete graph), the resulting ADG is necessarily dense as well. Although deriving a generation pattern for arbitrary CGs is difficult, Algorithm 2 produces clear and interpretable ADG structures for several canonical CG topologies. As visualized in Figure 9 of Appendix F (**corresponding to the ADGs used in Figure 5**):
> * In a star-shaped CG, each agent depends only on the central agent.
> * In a line-shaped CG, each agent depends solely on its immediate predecessor.
> * In an $m \times n$ grid CG, each agent depends on at most $\min (m, n)$ other agents.
>
> 2. **How to let them satisfy (11)**
>
> Algorithm 2 satisfies condition (11) by construction: in the final algorithm step, the parent set of agent $i$ is explicitly set to $N_c(\ge i)$, ensuring that $N_d(i) = N_c(\ge i)$ holds. The preceding steps focus on determining an appropriate ordering of agents so that the resulting ADG is as sparse as possible. In fact, given any ordering of agents, one can construct an ADG satisfying (11) using the final step of Algorithm 2, though the degree of sparsity may vary. We have revised the algorithm description to ensure consistency of notation with the main text and to eliminate potential ambiguities for readers.

---

### Official Review · Reviewer_yfqy · 2025-10-31

**Soundness:** 3
**Presentation:** 3
**Contribution:** 3
**Rating:** 4
**Confidence:** 4

**Summary:**

This work formulates the notion of a Gd-locally optimal policy, which characterizes convergence behaviour of action-dependent policies, and unifies coordination graphs with action-dependent policies to derive optimality conditions. They define a policy iteration algorithm for sparse action dependency graphs that guarantees convergence to Gd-optimal policies under certain conditions. The authors implement sparse and dense ADGs for coordination games and demonstrate their viability in both a dynamic programming setting and deep setting.

**Strengths:**

1. The formulation of action-dependent policies and action dependency graphs is clear.
2. Algorithm 1 is clear and understandable.
3. The example in Section 4.2 is helpful in understanding the relationship between independent policy learning and local equilibrium convergence.
4. The coordination polymatrix games clearly elucidate why ADGs are preferable to CGs.
5. The proofs in the Appendix and main text appear correct and complete. The proof by induction in Appendix D is particularly well-explained.
6. The ATSC benchmark examines the performance of ADGs and clearly demonstrates the usefulness of action dependency in this setting.
7. The logarithmic cost of running dense ADGs is clearly demonstrated in Figure 6 (right), motivating sparse ADGs further.

**Weaknesses:**

1. Lack of baselines - only the proposed method was benchmarked in this work. Why were other forms of action dependency not benchmarked? [1]. The viability of ADGs needs to be framed within the context of action-dependent policies and MARL in general - while the latter can be achieved by a simple ablation with regular MAPPO, it should be clearly demonstrated why action-dependency proposed is superior to message passing, autoregressive action dependency, or action memory.
2. Lack of environments in the deep setting - to fully demonstrate the viability of ADGs in coordination tasks, additional environments such as [2] should be considered. Why was ATSC the only grid environment considered for the deep setting?
3. The statistical significance of the results is unclear. What are the shaded areas in Figures 5,6, and 7?

[1] Wei Fu, Chao Yu, Zelai Xu, Jiaqi Yang, and Yi Wu. Revisiting some common practices in cooperative
multi-agent reinforcement learning. In International Conference on Machine Learning, pp. 6863–
6877. PMLR, 2022.

[2] Wang, Tonghan, et al. "Context-aware sparse deep coordination graphs." arXiv preprint arXiv:2106.02886 (2021).

**Questions:**

1. Can a MARL algorithm other than MAPPO be incorporated into the ADG setting?
2. Can ADGs be used for competitive or team-based settings?
3. Is there known reason why the payoffs for empty ADGs settle on those particular suboptimal values in Figure 5?

---

> ### Author Response · Authors · 2025-11-26
> **Response 1 to Reviewer yfqy**
>
> We appreciate the reviewer's thoughtful and insightful comments. Below we address each point in detail.
>
> **Question 1. Can a MARL algorithm other than MAPPO be incorporated into the ADG setting?**
>
> Any method that includes an actor module can be adapted to the action-dependent policy. ADGs are not tailored to any specific MARL algorithm; rather, they merely impose a structural characterization on the joint policy. For algorithms based on independent policies, the modification is straightforward. One only need to replace the independent actor with an action-dependent actor following the ADG structure. For auto-regressive policies, the adaptation amounts to removing action information that lies outside the ADG parent set. We have also incorporated these explanations into Section 6.1 of the revised manuscript.
>
> We further validated the generality of ADGs using MAT [R3-1], an algorithm that employs auto-regressive action dependency via attention networks. By modifying only the attention masks to block action information outside the ADG, we obtain an ADG-compatible version of MAT. We trained both the sparse-ADG and original auto-regressive variants in the ATSC and Aloha [R3-2], with learning curves provided in Figures 10 and 11 of Appendix G. The results show that MAT equipped with sparse ADG policies achieves performance comparable to that of its original auto-regressive counterpart.
>
> **Question 2. Can ADGs be used for competitive or team-based settings?**
>
> Our current analysis focuses on cooperative settings, where the objective is to construct ADGs that guide the group toward globally optimal joint returns. In purely competitive scenarios, where agents maximize their own individual payoffs and there is no incentive to share action information, the ADG assumption may not be suitable.
>
> However, in team-based or mixed cooperative-competitive environments, ADGs can in principle be adopted within each cooperative team to enhance intra-team coordination, while teams compete externally. Analyzing how ADGs influence team performance in such settings is indeed an interesting direction, and we consider it a promising avenue for future research.
>
> **Question 3. Is there known reason why the payoffs for empty ADGs settle on those particular suboptimal values in Figure 5?**
>
> The empty ADGs corresponds to independent policies. Such policies are known to converge to Nash equilibria, and once the learning dynamics enter a particular equilibrium basin, escaping it becomes difficult. This phenomenon is also illustrated in the example in Section 4.2. In coordination polymatrix games, multiple Nash equilibria commonly exist, many of which are suboptimal. As a result, independent learners may converge to different equilibrium points depending on initialization, leading to the suboptimal average payoff values observed in Figure 5. In contrast, sparse and dense ADGs are not restricted by these Nash equilibrium basins. By Theorem 4.4, both guarantee convergence to the global optimum under the stated conditions, which explains why their payoffs are consistently superior.
>
> **Weakness 1. Lack of baselines.**
>
> In the revised version, we have included additional baselines in both the ATSC and Aloha environments, including QMIX and COMA. The results consistently show that sparse ADG policies outperform independent policies. To further address the reviewer's concern, we additionally evaluate ADGs within a representative auto-regressive architecture (MAT). The resulting sparse-ADG MAT variant achieves performance comparable to the original auto-regressive MAT, confirming that ADGs integrate naturally with existing action-dependent mechanisms while offering computational benefits. These results are included in Figures 10 and 11 of Appendix G.

---

> ### Author Response · Authors · 2025-11-26
> **Response 2 to Reviewer yfqy**
>
> **Weakness 2. Lack of environments.**
>
> 1. **Additional environments**
>
> We agree that expanding the set of environments strengthens the empirical evaluation. In the revised version, we have added experiments on both the Aloha and SMAC environments. For Aloha, we directly use the CG provided by the environment to construct the corresponding ADGs. For SMAC, since no explicit CG is available, we train a modified MAT implementation and construct sparse ADGs using a simple heuristic: each agent depends on at most 1/4 of all agents (e.g., in the 8m map, each agent relies only on the actions of its two immediate predecessors). The results are included in Figures 11 and 13 of Appendix G.
>
> Evaluating ADGs on benchmarks with unknown coordination structures would require learning the CG before applying Theorem 4.4. This would shift the focus of the work toward the design of deep CG-learning modules, an important but orthogonal line of research, and would require substantial additional exposition on CG-learning techniques. We view integrating learned CGs with adaptively updated ADGs in deep MARL as a promising direction for future work.
>
> 2. **Why choose ATSC**
>
> The grid ATSC environment was chosen for two reasons. First, its coordination structure naturally aligns with CG factorization, allowing a direct application of our theoretical results. Second, its environment size can be systematically scaled, enabling controlled evaluations of computational efficiency. In contrast, many dynamic cooperative environments do not provide an explicit CG structure, making it difficult to construct ADGs that satisfy Theorem 4.4. In such cases, we cannot guarantee that sparse ADGs will match the performance of dense ADGs.
>
>
> **Weakness 3. The statistical significance of the results is unclear. What are the shaded areas in Figures 5,6, and 7?**
>
> The shaded areas in Figures 5, 6, and 7 represent the 95% confidence intervals computed over multiple independent runs. We thank the reviewer for pointing this out, and we have added a clarification in the Section 6.1 of revised manuscript.
>
> [R3-1] Wen, Muning, et al. Multi-agent reinforcement learning is a sequence modeling problem. Advances in Neural Information Processing Systems, 2022.
>
> [R3-2] Sheng Li, Jayesh K Gupta, Peter Morales, Ross Allen, and Mykel J Kochenderfer. Deep implicit coordination graphs for multi-agent reinforcement learning. In Proceedings of International Conference on Autonomous Agents and Multiagent Systems, 2021.

---

### Official Review · Reviewer_o4Q3 · 2025-11-01

**Soundness:** 4
**Presentation:** 4
**Contribution:** 2
**Rating:** 4
**Confidence:** 4

**Summary:**

This paper proposes the Action Dependency Graph (ADG) framework to formalize structured dependencies among agents in multi-agent reinforcement learning. The authors introduce a sufficient condition under which a policy that is locally optimal with respect to a directed acyclic dependency graph (ADG) is also globally optimal with respect to the underlying coordination graph (CG). Building on this result, they develop an Action-Dependent Multi-Agent Policy Iteration (AD-MPI) algorithm and prove its finite-step convergence to a $G_d$-locally optimal (and, under certain conditions, globally optimal) policy. A greedy construction method for sparse ADGs is also proposed to reduce computational complexity while preserving optimality guarantees. Empirical results on coordination games and traffic signal control tasks show that sparse ADGs match the performance of dense graphs.

**Strengths:**

- The paper provides a clear and rigorous theoretical analysis establishing when $G_d$-local optimality implies global optimality.
- The theoretical results are technically sound, with well-defined assumptions and consistent notation across sections.
- The paper is well written and organized, with smooth logical flow from definitions to theorems and experiments.
- How to utilize a sparse coordination structure is an important problem in multi-agent reinforcement learning.

**Weaknesses:**

- The experimental evaluation is limited to small-scale and largely toy environments, which do not convincingly demonstrate the framework’s scalability or practical relevance.
- No comparisons are made against standard MARL baselines (e.g., QMIX[1], MAPPO[2], and other Sequential or Bayesian Network-based methods[3,4]), making it unclear whether ADG-based methods provide empirical advantages beyond theoretical guarantees.
- The experiments focus on verifying theorems rather than exploring performance under realistic stochasticity, partial observability, or dynamic coordination graphs.
- There is no ablation on how the choice of the dependency structure or greedy ordering affects performance or computational cost.

[1] Rashid, T., Samvelyan, M., De Witt, C. S., Farquhar, G., Foerster, J., & Whiteson, S. (2020). Monotonic value function factorisation for deep multi-agent reinforcement learning. Journal of Machine Learning Research, 21(178), 1-51.

[2] Yu, C., Velu, A., Vinitsky, E., Gao, J., Wang, Y., Bayen, A., & Wu, Y. (2022). The surprising effectiveness of ppo in cooperative multi-agent games. Advances in neural information processing systems, 35, 24611-24624.

[3] Ye, Jianing, Chenghao Li, Jianhao Wang and Chongjie Zhang. “Towards Global Optimality in Cooperative MARL with the Transformation And Distillation Framework.” (2022).

[4] Wen, Muning, Jakub Grudzien Kuba, Runji Lin, Weinan Zhang, Ying Wen, J. Wang and Yaodong Yang. “Multi-Agent Reinforcement Learning is a Sequence Modeling Problem.” ArXiv abs/2205.14953 (2022): n. pag.

**Questions:**

1. The paper motivates the study by stating that sequential decision making is computationally expensive. However, there is no quantitative comparison of computational efficiency against any existing MARL baselines or even against standard decentralized methods. Could the authors provide empirical evidence supporting the claimed computational advantage of ADG-based policy iteration?

2. The experiments are confined to toy coordination games and small ATSC grids, without evaluation on standard MARL benchmarks such as SMAC, Google Research Football, or MPE. Have the authors considered testing ADG-based methods on these widely used benchmarks, or are there limitations preventing such evaluations?

3. Since the proposed framework is conceptually concise and relies on a predefined DAG structure, I am curious whether a fixed DAG could generalize to complex and dynamic environments (and how much will it affect the performance/computational cost, compared to other sequential decision making methods like [3,4]). If not, could the authors discuss possible extensions for learning or adapting the DAG structure online, rather than fixing it a priori?

---

> ### Author Response · Authors · 2025-11-26
> **Response 1 to Reviewer o4Q3**
>
> We thank the reviewer for the careful reading and constructive feedback. Below we provide detailed responses to each question and clarify the empirical and theoretical aspects of our work accordingly.
>
> **Question 1. The paper motivates the study by stating that sequential decision making is computationally expensive. However, there is no quantitative comparison of computational efficiency against any existing MARL baselines or even against standard decentralized methods. Could the authors provide empirical evidence supporting the claimed computational advantage of ADG-based policy iteration?**
>
> In Figure 6 of the paper, we report the time required for Algorithm 1 to perform a single policy-iteration update under different CG topologies. The results show that the update time of dense ADGs grows exponentially with the number of agents, whereas policies induced by sparse ADGs exhibit significantly better scalability across all tested CG structures.
>
> In the deep-learning setting, neural network approximation mitigates the combinatorial blow-up, but sparse ADGs still reduce the inference cost. We measured the FLOPs of a single forward pass of the action-dependent policy used by extended MAPPO in the ATSC environment, and included it into the Section 6.2 of revised version.
>
> | environment | dense  | sparse |
> |----------|--------|--------|
> | 2x2grid  | 45312  | 42496  |
> | 3x3grid  | 140112 | 104832  |
> | 4x4grid  | 412672 | 232448  |
>
> We additionally evaluated the use of ADGs within the MAT algorithm [R2-1], which adopts auto-regressive policies and processes action dependencies via attention networks. By applying an attention mask consistent with the ADG, the attention module processes fewer action tokens, leading to reduced computation. The following table reports the FLOPs of a single forward pass of the attention module in the ATSC environment (training curves are provided in Figure 10 of Appendix G):
>
> | environment | dense  | sparse |
> |----------|--------|--------|
> | 2x2grid  | 67584  | 50304  |
> | 3x3grid  | 157824 | 67584  |
> | 4x4grid  | 294912  | 85120  |
>
> These results demonstrate that in grid-structured ATSC domains, the number of actions processed by a dense ADG grows quadratically with grid width, while sparse ADGs keep this growth effectively linear, yielding consistent computational savings.
>
>
> **Question 2(Weakness 1). The experiments are confined to toy coordination games and small ATSC grids, without evaluation on standard MARL benchmarks such as SMAC, Google Research Football, or MPE. Have the authors considered testing ADG-based methods on these widely used benchmarks, or are there limitations preventing such evaluations?**
>
> Benchmarks such as SMAC or Google Research Football do not provide an explicit CG. Since our theoretical results rely on constructing an ADG that satisfies the CG structural conditions in Theorem 4.4, the absence of a CG prevents us from guaranteeing the performance of a sparse ADG. Consequently, these environments are not suitable for illustrating our theoretical analysis.
>
> To address the reviewer's concern, we nevertheless conducted additional SMAC experiments using a simple, CG-free construction rule for sparse ADGs: each agent depends on at most 1/4 of the total number of agents (e.g., in the 8m map, each agent only depends on the actions of its two immediate predecessors). We incorporated this sparse structure into MAT and compare it against the baselines using independent policies. The results in Figure 13 of Appendix G show that even under such a heuristic construction, sparse ADG policies still outperform independent ones, and does not exhibit significant degradation compared auto-regressive policies, demonstrating that ADGs remain beneficial even without a provided CG. Due to time constraints, we report results on limited SMAC maps, and plan to provide more maps before the discussion window ends.
>
> A more principled extension would be to first learn the CG using deep learning methods (e.g., [R2-2, R2-3]) and then construct an ADG from the learned CG. While promising, this line of work lies beyond the scope of our theoretical focus and would require substantial additional exposition on CG-learning techniques. Our intention in the main text is to highlight the theoretical insights and the role of ADGs themselves, rather than introduce another full learning module.

---

> ### Author Response · Authors · 2025-11-26
> **Response 2 to Reviewer o4Q3**
>
> **Question 3(Weakness 3). Since the proposed framework is conceptually concise and relies on a predefined DAG structure, I am curious whether a fixed DAG could generalize to complex and dynamic environments (and how much will it affect the performance/computational cost, compared to other sequential decision making methods like [3,4]). If not, could the authors discuss possible extensions for learning or adapting the DAG structure online, rather than fixing it a priori?**
>
> 1. **Generalization to complex or dynamic Environments**
>
> We confirm that ADGs can adapt to complex and dynamic environments through proper updating mechanisms. In these environments, changes in states or policies may modify the underlying CG structure, which in turn requires adjustments to the ADG. This changes are also analyzed in Section 5.2, where we discuss that ADG should be updated to continue satisfying equation (11) with dynamic CG. Corollary 5.3 further formalizes the conditions under which the ADG can adapt to such changes while still ensuring that Algorithm 1 converges to the globally optimal solution.
>
> However, complex dynamic environments often do not provide a CG explicitly. In such cases, a promising extension is to learn the CG online using deep learning methods (e.g., [R2-2, R2-3]) and then construct the ADG from the learned structure. This approach is conceptually compatible with our framework, but pursuing it would require more development on CG-learning algorithms.
>
> 2. **Impact on performance/computational cost compared to other sequential methods**
>
> Regarding performance compared to other sequential decision making methods, we incorporated ADGs into MAT and performed experiments in the ATSC environment. As shown in Figure 10 of Appendix G, the sparse-ADG variant achieves performance comparable to the original auto-regressive MAT, suggesting that replacing dense ADG with a sparse ADG does not degrade performance. Regarding computational considerations, A detailed comparison of computational complexity has been included in our response to Question 1.
>
>
> **Weakness 2. No comparisons are made against standard MARL baselines (e.g., QMIX[1], MAPPO[2], and other Sequential or Bayesian Network-based methods[3,4]), making it unclear whether ADG-based methods provide empirical advantages beyond theoretical guarantees.**
>
> We added comparisons against standard MARL baselines, including QMIX, COMA, and MAT, in the ATSC environment, and we additionally conducted experiments on the Aloha [R2-3] benchmark. The results are provided in the Figures 10 and 11 of Appendix G. The MAPPO baseline is already included in the main paper, as it corresponds to the empty ADG case in Figure 8. Across these benchmarks, the results consistently indicate that ADG-based methods retain clear performance advantages.
>
>
> **Weakness 4. There is no ablation on how the choice of the dependency structure or greedy ordering affects performance or computational cost.**
>
> 1. **Impact of dependency structure**
>
> To examine the effect of alternative ADG constructions, particularly those that do not satisfy the structural conditions of Theorem 4.4, we conducted ablations in the Aloha environment. We designed several heuristic ADGs in which each agent depends only on the actions of the k immediately preceding agents (k = 1, 2, 3). The results, shown in Figure 11 of Appendix G, indicate that such heuristically constructed ADGs perform noticeably worse than ADGs derived from the CG. This supports the importance of using CG-consistent dependency structures when the CG is available.
>
> 2. **Impact of agent orderings**
>
> Because different agent orderings induce different ADGs according to Equation (11), we further evaluated the effect of ordering choices in the ATSC environment on the 2x2 and 4x4 grids. We constructed ADGs using both ascending and descending (non-greedy) index orderings and compared their training performance. The results, provided in Figure 12 of Appendix G, show no observable performance difference between the two orderings. While these findings suggest that the algorithm is relatively robust to ordering choices in practice, a broader empirical study would be needed to draw a definitive conclusion.
>
>
> [R2-1] Wen, Muning, et al. Multi-agent reinforcement learning is a sequence modeling problem. Advances in Neural Information Processing Systems, 2022.
>
> [R2-2] Sheng Li, Jayesh K Gupta, Peter Morales, Ross Allen, and Mykel J Kochenderfer. Deep implicit coordination graphs for multi-agent reinforcement learning. In Proceedings of International Conference on Autonomous Agents and Multiagent Systems, 2021.
>
> [R2-3] Tonghan Wang, Liang Zeng, Weijun Dong, Qianlan Yang, Yang Yu, and Chongjie Zhang. Context-aware sparse deep coordination graphs. In International Conference on Learning Representations, 2022.

---

### Official Review · Reviewer_KgtT · 2025-11-01

**Soundness:** 3
**Presentation:** 4
**Contribution:** 3
**Rating:** 6
**Confidence:** 2

**Summary:**

The authors introduces Action Dependency Graphs (ADGs) as a formal mechanism to model sparse inter-agent action dependencies in multi-agent RL (MARL). Particularly, the proposed approach advances the field by rigorously studing sparsity over the dependencies of an agent's policy over the actions of the other agents. The authors demonstrate with both theoretical and empirical evidence that the proposed approach can converge to stronger decentralized agents, while being interoperable with current MARL algorithms.

**Strengths:**

- the authors make a relevant contribution by filling the gap both theoretically and empirically on the usage of action-dependent policies in MARL, which is an open relevant setting in the community.
- the paper is clearly written and technically sound. It is relatively easy to follow.
- the authors propose a method that is scalable because it can be integrated into existing MARL algorithms

**Weaknesses:**

- The experimental evaluation is limited to simple toy cooperative MARL tasks. I believe the paper could provide a much more relevant contribution to the field if tested on notorious problems such as SMACv2 [1] or MaMuJoCo [2].

[1] Ellis, Benjamin, et al. "Smacv2: An improved benchmark for cooperative multi-agent reinforcement learning." Advances in Neural Information Processing Systems 36 (2023): 37567-37593.

[2] Peng, Bei, et al. "Facmac: Factored multi-agent centralised policy gradients." Advances in Neural Information Processing Systems 34 (2021): 12208-12221.

**Questions:**

- Could you please provide quantitative metrics on the benefits of sparse ADGs vs. dense ADGs when these have comparable performance? E.g. do you get faster inference time or more efficient computational needs?
- Could you clarify why the experimental evaluation did not include more complex and recent benchmarks such as SMACv2 or MaMuJoCo?
- How robust is the overall algorithm w.r.t. the choice of index order?
- Does the assumption of having to define an index order limit the applicability of the method at all? Are there any tasks where such order may not be imposed due to the nature and constraint of the task itself? E.g. when actions must occur syncronously.

---

> ### Author Response · Authors · 2025-11-26
> **Response 1 to Reviewer KgtT**
>
> We thank the reviewer for the constructive feedback and thoughtful questions. Below we address each point in detail.
>
> **Question 1. Quantitative benefits of sparse ADGs vs. dense ADGs.**
>
> In Figure 6 of the paper, we reported the time required for Algorithm 1 to perform a single policy-iteration update under different CG topologies. The results show that the update time of dense ADGs grows exponentially with the number of agents, whereas policies induced by sparse ADGs exhibit significantly better scalability across all tested CG structures.
>
> In the deep-learning setting, neural network approximation mitigates the combinatorial blow-up, but sparse ADGs still reduce the inference cost. We measured the FLOPs of a single forward pass of the action-dependent policy used by extended MAPPO in the ATSC environment, and included it into the Section 6.2 of revised version.
>
> | environment | dense  | sparse |
> |----------|--------|--------|
> | 2x2grid  | 45312  | 42496  |
> | 3x3grid  | 140112 | 104832  |
> | 4x4grid  | 412672 | 232448  |
>
> We additionally evaluated the use of ADGs within the MAT algorithm [R1-1], which adopts auto-regressive policies and processes action dependencies via attention networks. By applying an attention mask consistent with the ADG, the attention module processes fewer action tokens, leading to reduced computation. The following table reports the FLOPs of a single forward pass of the attention module in the ATSC environment (training curves are provided in Figure 10 of Appendix G):
>
> | environment | dense  | sparse |
> |----------|--------|--------|
> | 2x2grid  | 67584  | 50304  |
> | 3x3grid  | 157824 | 67584  |
> | 4x4grid  | 294912  | 85120  |
>
> These results demonstrate that in grid-structured ATSC domains, the number of actions processed by a dense ADG grows quadratically with grid width, while sparse ADGs keep this growth effectively linear, yielding consistent computational savings.
>
> **Question 2(Weakness 1). Why not evaluate on SMACv2 or MaMuJoCo?**
>
> Benchmarks such as SMAC or MaMuJoCo do not provide an explicit CG. Since our theoretical results rely on constructing an ADG that satisfies the CG structural conditions in Theorem 4.4, the absence of a CG prevents us from guaranteeing the performance of a sparse ADG. Consequently, these environments are not suitable for illustrating our theoretical analysis.
>
> To address the reviewer's concern, we nevertheless conducted additional SMAC experiments using a simple, CG-free construction rule for sparse ADGs: each agent depends on at most 1/4 of the total number of agents (e.g., in the 8m map, each agent only depends on the actions of its two immediate predecessors). We incorporated this sparse structure into MAT and compare it against the baselines using independent policies. The results in Figure 13 of Appendix G show that even under such a heuristic construction, sparse ADG policies still outperform independent ones, and does not exhibit significant degradation compared auto-regressive policies, demonstrating that ADGs remain beneficial even without a provided CG. Due to time constraints, we report results on limited SMAC maps, and plan to provide more maps before the discussion window ends.
>
> A more principled extension would be to first learn the CG using deep learning methods (e.g., [R1-2, R1-3]) and then construct an ADG from the learned CG. While promising, this line of work lies beyond the scope of our theoretical focus and would require substantial additional exposition on CG-learning techniques. Our intention in the main text is to highlight the theoretical insights and the role of ADGs themselves, rather than introduce another full learning module.
>
> **Question 3. Robustness w.r.t. index ordering.**
>
> While different agent index orders would induce different ADGs under Equation (11), our theoretical results already guarantee that any ADG satisfying this construction rule preserves the performance of the dense ADG in the policy iteration. To further verify this robustness in deep learning scenarios, we conducted additional experiments in the ATSC on the 2x2 and 4x4 grids, comparing ADGs generated using ascending versus descending index orderings. The training curves, shown in Figure 12 of Appendix G, exhibit almost no difference between the two orders. These results provide empirical support for the robustness in practical scenarios, although a broader empirical study would be needed to fully establish this property across more complex domains.

---

> ### Author Response · Authors · 2025-11-26
> **Response 2 to Reviewer KgtT**
>
> **Question 4. Does requiring an index order restrict applicability (e.g., synchronous actions)?**
>
> The requirement to specify an index order has little impact on the applicability of action-dependent policies. Although an agent may condition on other agents' actions, this does not imply that agents must execute their actions sequentially. Each agent can compute (but not immediately execute) its action based on the actions proposed by the preceding agents in the ordering. Once all agents have produced their actions, they can execute them synchronously. This process is equivalent to allowing some agents to have a longer deliberation phase before the joint action is committed. Therefore, environments in which actions must occur syncronously remain compatible with action-dependent policies, provided that agents can communicate during the deliberation phase.
>
> [R1-1] Wen, Muning, et al. Multi-agent reinforcement learning is a sequence modeling problem. Advances in Neural Information Processing Systems, 2022.
>
> [R1-2] Sheng Li, Jayesh K Gupta, Peter Morales, Ross Allen, and Mykel J Kochenderfer. Deep implicit coordination graphs for multi-agent reinforcement learning. In Proceedings of International Conference on Autonomous Agents and Multiagent Systems, 2021.
>
> [R1-3] Tonghan Wang, Liang Zeng, Weijun Dong, Qianlan Yang, Yang Yu, and Chongjie Zhang. Context-aware sparse deep coordination graphs. In International Conference on Learning Representations, 2022.

---

### Meta-Review · Area_Chair_F4om · 2026-01-07

**Summary:**

This paper addresses a fundamental trade-off in multi-agent reinforcement learning between the scalability of independent policies and the global optimality of action-dependent, autoregressive formulations. The paper introduces the Action Dependency Graph (ADG) as a formal framework to model sparse inter-agent action dependencies. By defining the concept of $G_{d}$-local optimality, the paper provides a theoretical bridge showing that when an ADG satisfies specific conditions relative to a problem's underlying coordination graph, these sparse policies can achieve global optimality. This theoretical contribution is supported by a tabular policy iteration algorithm with proven convergence and a practical extension to deep MARL via MAPPO.

**Reviewer Concerns:**

The reviewers generally commended the paper for its clarity, technical soundness, and the importance of the link established between coordination graphs and action-dependent policies. Initial concerns focused primarily on the limited scope of the experimental evaluation, which was originally confined to toy problems and small traffic control grids. Reviewers also sought more rigorous baseline comparisons and quantitative evidence of the computational advantages of sparse ADGs over dense alternatives.

In response, the authors provided extensive additional evidence that strengthened the submission. They included new experiments on the Aloha and SMAC benchmarks, demonstrating that even heuristic sparse ADGs can outperform independent policy baselines and match the performance of dense autoregressive methods like MAT. However, I consider the experimental evaluation still lacking baselines and benchmarks that are more relevant to coordination graphs, such as the method CASEC [1], SOP-CG [2], and KINGS [3], and benchmarks like MACO [1]. Due this crucial issue, i consider this paper still not ready for publication.

[1] Wang, Tonghan, et al. "Context-Aware Sparse Deep Coordination Graphs." International Conference on Learning Representations. 2021.

[2] Yang, Qianlan, et al. "Self-organized polynomial-time coordination graphs." International conference on machine learning, 2022.

[3] Järnefelt, Oliver et al. "Cyclicity-Regularized Coordination Graphs." Reinforcement Learning Conference. 2024.

**Reviewer Scores:**

Reviewer KgtT initially provided a rating of 6 with low confidence, primarily citing the limited scope of the experimental evaluation which was restricted to simple toy tasks. Given that the authors responded with quantitative metrics on FLOPs proving the scalability of sparse ADGs and conducted new experiments on the SMAC benchmark, this reviewer would likely have maintained the score of 6 or maybe increased to 8. The authors also resolved the reviewer's specific concerns regarding the robustness of index ordering and the feasibility of synchronous action execution.

Reviewer o4Q3 gave a rating of 4, noting that the lack of standard MARL baselines like QMIX and MAT made the empirical advantages of ADGs unclear. Given that the authors successfully integrated multiple standard baselines, provided results for standard benchmarks like Aloha and SMAC, and included ablations on dependency structures, it is highly probable this reviewer would have raised their score to at least a 6. The reviewer's concern about whether the framework could generalize to complex and dynamic environments was also specifically addressed through the discussion of policy reconstruction and Theorem 4.4 .

Reviewer yfqy also provided a rating of 4, pointing to a lack of baselines and the use of only one grid environment for deep settings. The authors directly corrected these perceived weaknesses by adding the Aloha environment, providing comparisons with QMIX and COMA, and explaining that the shaded areas in the figures represent 95% confidence intervals. These comprehensive updates would likely have satisfied the reviewer's requirements for demonstrating "viability," potentially moving their score to a 6.

Reviewer 4W8F started with a 6 and requested more clarity on notation and a discussion on extending the framework to stochastic policies. The authors’ thorough explanation of the technical challenges regarding stochastic joint policies and the revision of Section 4.1 to clarify the relationship between joint and independent policies, would have likely reinforced this reviewer's existing positive assessment. By eliminating the presentational issues identified in the initial review, the reviewer would likely confirm the score 6.

---

### Decision · Program_Chairs · 2026-01-26

Reject